# Late Miocene to Recent High Resolution Eastern Equatorial Pacific Carbonate Records: Stratigraphy linked by dissolution and paleoproductivity

Mitchell Lyle[1], Anna Joy Drury[2,5], Jun Tian[3], Roy Wilkens[4], and Thomas Westerhold[2]

[1]College of Earth, Ocean, and Atmospheric Science, Oregon State University, 104 CEOAS Admin Bldg, Corvallis, Oregon 97331, USA

[2]MARUM-Center for Marine Environmental Sciences, University of Bremen, Leobener Strasse, DE-28359 Bremen, Germany

[3] Laboratory of Marine Geology, Tongji University, Siping Road 1239, Shanghai 200092, PR China

[4] University of Hawaii, School of Ocean and Earth Science and Technology, Honolulu Hawaii 96822, USA

[5] now at University College London, Earth Sciences, London WC1E 6BS, UK

*Correspondence to*: Mitchell Lyle (mlyle@coas.oregonstate.edu)

**Abstract.** Coherent variation of $CaCO_3$ burial is a feature of the Cenozoic eastern equatorial Pacific. Nevertheless, there has been a long-standing ambiguity whether changes in $CaCO_3$ dissolution or changes in equatorial primary production might cause the variability. Since productivity and dissolution leave distinctive regional signals, a regional synthesis of data using updated age models and high-resolution stratigraphic correlation is an important constraint to distinguish between dissolution and production as factors that cause low $CaCO_3$. Furthermore the new chronostratigraphy is an important foundation for future paleoceanographic studies. The ability to distinguish between primary production and dissolution is also important to establish a regional carbonate compensation depth (CCD). We report late Miocene to recent time series of X-ray Fluorescence (XRF) derived bulk sediment composition and *mass accumulation rates* (MAR) from eastern equatorial Pacific Integrated Ocean Drilling Program (IODP) Sites U1335, U1337, U1338 and Ocean Drilling Program (ODP) Site 849, and also report bulk density derived $CaCO_3$ MAR at ODP Sites 848, 850 and 851. We use physical properties, XRF bulk chemical scans, and images along with available chronostratrigraphy to inter-correlate records in depth space. We then apply a new equatorial Pacific age model to create correlated age records for the last 8 Myr with resolutions of 1-2 kyr. Large magnitude changes in $CaCO_3$ and bio-$SiO_2$ (biogenic opal) MAR occurred within that time period but clay deposition has remained relatively constant, indicating that changes in Fe deposition from dust is only a secondary feedback to equatorial productivity. Because clay deposition is relatively constant, ratios of $CaCO_3$ % or biogenic $SiO_2$ % to clay emulate changes of biogenic MAR. We define 5 major *Plio-Pleistocene Low CaCO₃ %* (PPLC) intervals since 5.3 Ma. Two were caused primarily by high bio-$SiO_2$ burial that diluted $CaCO_3$ (PPLC-2—1685-2135 ka, and PPLC-5—4465-4737 ka), while 3 were caused by enhanced dissolution of $CaCO_3$ (PPLC-1—51-402 ka, PPLC-3—2248-2684 ka, and PPLC-4—2915-4093 ka). Regional patterns of $CaCO_3$ % minima can distinguish between low $CaCO_3$ caused by high diatom bio-$SiO_2$ dilution versus lows caused by high $CaCO_3$ dissolution. $CaCO_3$ dissolution can be confirmed through scanning XRF measurements of Ba.

High diatom production causes lowest $CaCO_3$ % within the equatorial high productivity zone, while higher dissolution causes lowest $CaCO_3$ at higher latitudes where $CaCO_3$ production is lower. The two diatom production intervals, PPLC-2 and PPLC-5, have different geographic footprints from each other because of regional changes in eastern Pacific nutrient storage after the closure of the Panama Seaway. Because of the regional variability in carbonate production and sedimentation, the carbonate compensation depth (CCD) approach is only useful to examine large changes in $CaCO_3$ dissolution.

## 1 Introduction

IODP Expeditions 320/321 drilled 8 sites in the tropical Pacific in 2009. One of the primary objectives of IODP Expeditions 320/321 (PEAT, Pacific Equatorial Age Transect; Lyle et al, 2010) was to join the equatorial portions of essentially continuous sediment records from the 8 PEAT drill sites and to intercorrelate the PEAT records with ODP Leg 138 drilling for the late Neogene (Mayer et al, 1992) to study the evolution of the eastern equatorial Pacific for the last 50 million years. In addition, new chronostratigraphy from the PEAT drilling (e.g., Channell et al, 2013; Holbourn et al, 2014; Kochann et al, 2016; Drury et al, 2017, 2018) would be used to refine and update equatorial Pacific age models. Another important objective from the PEAT expeditions was to identify major changes and causes of carbonate storage in the eastern equatorial Pacific. Pälike et al (2012) did a synthesis of $CaCO_3$ data on a Cenozoic scale to identify changes in the CCD in the equatorial Pacific using data from the PEAT expeditions and earlier drilling, but new studies are needed to add detail to the record.

### 1.1 Objectives: CaCO3 deposition through site to site correlation of the late Miocene-Recent sections from Exp 321 and ODP Leg 138

We develop a regional site-to-site correlation between 7 drill sites in the eastern Pacific and apply a new late Miocene-Recent age model in order to study $CaCO_3$ production and dissolution in the eastern Pacific. The regional CCD has been relatively deep since the middle Miocene and drill sites have not intersected the local CCD. In order to understand causes of eastern Pacific $CaCO_3$ variability we re-examine the Pliocene and Pleistocene interval, to distinguish intervals of $CaCO_3$ dissolution and productivity in a time interval and region where the CCD is typically deeper than the eastern Pacific sea floor. We explore alternate methods of developing the CCD by extrapolating $CaCO_3$ MARs from drill sites above the CCD to the depth of zero burial (Lyle et al., 2005) and use XRF-derived $CaCO_3/BaSO_4$ as measure of local $CaCO_3$ dissolution (Lyle and Baldauf, 2015). We examine regional variation in the 7 eastern Pacific sites where we have made high-resolution depth correlations and assigned ages with a new eastern Pacific age model and new stable isotope records (Drury et al, 2016, 2017, 2018; Tian et al., 2018).

A key aim is to disentangle $CaCO_3$ variability caused by dissolution, versus high bio-$SiO_2$ paleoproductivity using bulk sediment chemistry and regional patterns of variation. Production intervals have a different $CaCO_3$ expression at different drill sites because of regional differences in relative nannofossil production of $CaCO_3$ versus diatom production of bio-$SiO_2$. We will primarily discuss data for the interval from 5.3-0 Ma although the 7 eastern Pacific sites are depth correlated to $\geq 10$ Ma and the age model has been applied to ~8 Ma for all sites (see Supplemental Material). We will briefly discuss the interval between 8 and 4.5 Ma (the Late Miocene Biogenic Bloom, LMBB) because it has significantly more variability among the drill sites and needs further work. This depositional variability has been imposed in the eastern Pacific by the regional footprint of paleoproductivity within the LMBB interval.

## 1.2 Background: Cenozoic CaCO$_3$ deposition

Cenozoic sediments of the eastern tropical Pacific Ocean have recorded changes to productivity and to the Pacific deep carbon reservoir (Berger, 1973; van Andel et al, 1975; Pälike et al, 2012). Distinctive $CaCO_3$ profiles can be correlated across thousands of km within the eastern equatorial Pacific because of coherent $CaCO_3$ variability within the region (Mayer et al, 1986). The large geographic response to ENSO (el Niño-Southern Oscillation; Mantua et al, 1997; Di Lorenzo et al, 2015) demonstrates that the whole surface eastern tropical Pacific changes coherently on the interannual time scale. Large-scale flow of abyssal waters southward along the western flank of the East Pacific Rise (Johnson and Toole, 1993; Hautala, 2018) causes the western flank to be bathed by a common bottom water and to share a common signal of abyssal change. Both surface and abyssal processes interact to drive these common $CaCO_3$ profiles across the equatorial region. Since $CaCO_3$ variability causes large changes of sediment bulk density in the eastern tropical Pacific (Mayer, 1979, Mayer, 1991; Reghellin et al., 2013), the extensive geographic scale of coherent $CaCO_3$ profiles produces a distinctive set of seismic reflection horizons (Bloomer and Mayer, 1997; Tominaga et al., 2011) linked with major paleoceanographic change in the Pacific Ocean (Mayer et al, 1986).

Reconnaissance paleoceanographic descriptions of the Cenozoic equatorial Pacific (van Andel et al, 1975; Berger, 1973) often gloss over the late Miocene and early Pliocene as being periods almost like the present. The CCD variability within the last 12 million years is not nearly as dramatic as the huge changes earlier in the Cenozoic (Pälike et al, 2012). Nevertheless, as more detail has been generated from better chronology and more comprehensive regional records, significant changes in carbonate deposition can be found throughout the equatorial Pacific since the beginning of the late Miocene. Variability in the late Miocene and afterward represent profound changes in how $CaCO_3$ has been produced and recycled within the Pacific.

The change in the calcite compensation depth (CCD) in the world's oceans is a primary indicator of rearrangement of the global carbon cycle. Well-known major changes in CCD are associated with the major climate changes of the Cenozoic (Berger, 1973; van Andel and Moore, 1974; Peterson et al., 1992; Pälike et al., 2012). Once sediments are buried, pore water

concentrations quickly reach saturation with respect to $CaCO_3$. Buried $CaCO_3$ is unaffected by subsequent changes in ocean chemistry, so the variation reflects the state of the ocean at the time of burial. Fluctuations in the CCD represent changes in the balance between $CaCO_3$ production in surface waters and $CaCO_3$ dissolution of carbonate hard parts at the benthic boundary layer (Milliman and Droxler, 1996; Boudreau et al., 2010). Surface production of $CaCO_3$ has a large effect upon the CCD as shown by the CCD being 1 km deeper underneath the equatorial Pacific zone of high productivity versus on its flanks (Berger et al., 1973; van Andel and Moore, 1974).

The CCD shallows when (a) lower carbonate production results in lower $CaCO_3$ rain to the seafloor, (b) the $[CO_3]^{2-}$ content of bottom water decreases and becomes undersaturated with respect to calcite, or (c) rising sea levels allow significantly more $CaCO_3$ storage on continental shelves and shallow basins (Opdyke and Wilkinson, 1988). Because the CCD strongly responds to changes in production and regional rates of $CaCO_3$ burial, the global CCD has a complex response to changes in Cenozoic climate. The CCD differs between ocean basins because ocean circulation can alter where $CaCO_3$-undersaturated water impinges on sediments regionally (Berger, 1970). The Holocene deep Atlantic Ocean is flushed with deep water with high $[CO_3]^{2-}$ content originating from the surface North Atlantic. Higher $CaCO_3$ saturation results in a much deeper CCD than that in the Pacific. The Atlantic CCD shallows in glacial intervals because of the loss of North Atlantic Deep Water (NADW) and intrusion of more undersaturated water from the Antarctic, while the Pacific CCD deepens as the intrusion of low $[CO_3]^{2-}$ water into the Atlantic basin dissolves $CaCO_3$ there rather than in the Pacific (Crowley, 1985). The causes for CCD change can be disentangled but require sedimentary records from the different ocean basins as well as additional bulk chemical information beyond $CaCO_3$ %.

The CCD has varied both on glacial-interglacial scales and over long periods of time. Since large areas of the abyssal ocean basins are subject to $CaCO_3$ dissolution, climate variability imposes a common regional $CaCO_3$ stratigraphy modulated by the change in factors that perturb rates of $CaCO_3$ burial. Such a common chronostratigraphic signal in the eastern equatorial Pacific has been noted through the Cenozoic (Hays et al, 1969; Mayer et al, 1986; Chuey et al, 1987; Farrell and Prell, 1989; Farrell and Prell, 1991; Hagelburg et al., 1995). The development of the typical eastern Pacific $CaCO_3$ burial profile is complicated by large-scale long-term variation in biogenic production (Farrell et al, 1995; Lyle and Baldauf, 2015). Such variability can change the dissolution-related $CaCO_3$ signal by changing not only the mass of $CaCO_3$ that rains to the sea floor but also the relative proportion of $CaCO_3$ to biogenic opal (bio-$SiO_2$) in the rain of biogenic hard parts. Nevertheless, knowledge of the processes that modify $CaCO_3$ deposition allows for common intervals to be identified.

## 2 Drill Sites Studied

We report data from 7 eastern equatorial Pacific drill sites, 4 with XRF scans along the sediment splice (IODP Sites U1335, U1337, U1338, ODP Site 849) and 4 with $CaCO_3$ data estimated from spliced Gamma Ray Attenuation (GRA) wet bulk

density profiles (ODP Sites 848, 849, 850 and 851; see Methods for the process). Locations for the drill sites are shown in Figure 1 and in Table 1.

The 7 sites span the equatorial Pacific productivity zone and their positions range from 3°S to 5.3°N (Table 1). At 5 Ma, the sites positions ranged from ~4°S to ~4°N based upon a fixed hotspot backtrack. All of the sites are west of the East Pacific Rise and contain sediments that are primarily biogenic in origin. Sites that are further from the equator and/or deeper have more clay in the sediments (Sites U1335, U1337, and 851) than those shallower and nearer the equator, primarily because of lower carbonate production and preservation. Higher rainout of dust near the ITCZ, located north of the equator, results in higher dust fluxes and higher clay percentages in the north as well (Hovan, 1995).

The ODP Leg 138 drillsites (Sites 848-851) are positioned between 3°S to 2.8°N along 110°W (Mayer et al., 1992). They are correlated with Neogene sites from IODP Expedition 320/321 further west. All sites reside on the Pacific tectonic plate and have been translated to the NW over time by Pacific plate movement (Pälike et al, 2010). The Leg 138 sites have been cored to basement, which was formed about 11 Ma along 110°W. The Exp 320/321 sites cored to progressively older basement by following a common tectonic flow line from the east Pacific Rise. Each Expedition 320/321 drill site preserves a sediment record of progressively older Pacific equatorial regions. Because they are on older ocean crust than the late Miocene basement recovered at ODP Leg 138 drill sites, the Exp 320/321 drill sites have sediments that extend back to the middle/lower Miocene boundary at Site U1338, to the Oligocene/Miocene boundary at Site U1337, and to the late Oligocene at Site U1335 (Pälike et al, 2010).

Primarily because of depth dependent carbonate dissolution, Site U1338 has a sedimentation rate only 75% of that at Site 851, which lies at the same latitude but 450 m shallower. The depth difference is because Site 851 is situated on ocean crust that is half the age of Site U1338. From 5.3 Ma to the Holocene each site deepened by about 200 m because of crustal cooling partly mitigated by sediment deposition that shallowed the sea floor. Isostatic subsidence due to sediment loading increased the depth somewhat but subsidence is only ~20% of the added sediment thickness (Lyle, 1997). While significant changes in depth have happened to the 7 drill sites because of crustal cooling and sedimentation, the sites always remained separated by a similar water depth.

### 3 Methods

Please see the supplemental material for more information about the assembly and calibration of data, including the age model, stratigraphic tie points, revised splices, estimated CaCO$_3$% profiles, and XRF bulk chemical data.

### 3.1 Stratigraphic Correlation of eastern Pacific drill sites

We used the Code for Ocean Drilling Data (CODD; https://www.codd-home.net/) analytical suite of software tools described by Wilkens et al (2017) for the stratigraphic correlation of all the drillsites. We first investigated all the offsets and splice tie points at each multi-hole drill site to check that the ties were correct. We then built a continuous sediment section for each drill site by linking the sections within the splice. The spliced sediment columns were used to make site-to-site depth correlations. Descriptions of the splicing are given in the Supplemental material and splices used are in Supplemental Tables SM-1 to SM-7. Only Site 849 had significant changes from the shipboard splice, in the lower section older than 4 Ma.

Initial site-to-site correlations were made with the shipboard physical properties data from ODP or IODP collected by scans of the unopened cores prior to splitting, primarily magnetic susceptibility and GRA bulk density. If XRF and stable isotope data were available, the site-to-site correlations were refined with the additional information. During site-to-site correlation, the spliced sediment section at each drill site was stretched/squeezed to match the splice of a master drill site so that the sediment section could be expressed in the equivalent depth of the master site. Site U1338 was chosen as the master drill site, as it has the best balance between high sedimentation rates and sediment recovery across the last 8 Myr. It is also suitably situated in the centre of the studied sites. The CODD analytical suite allows age datum levels like magnetochrons and biostratigraphic boundaries to be displayed in the correlation space, and these were used to constrain the depth matching. Depth ties between drill sites can be found in Supplemental Tables SM-8 to SM-13

Correlations were achieved as equivalent depth at Site U1338. Sites U1335, U1337, 851, and 849 were tied directly to Site U1338, while Sites 848 and 850 were first tied to Site 849, and then correlated to Site U1338 through the Site 849 assigned U1338 equivalent depth. We found that there was less ambiguity to work in this fashion because of regional differences in the $CaCO_3$ profiles. Latitudinal differences in $CaCO_3$ production make nearby records easier to correlate. For this reason, Site 851 was directly correlated to Site U1338 because it was at the same latitude and its record was more similar to Site U1338 than Site 849. Site 850 has very similar profiles to Site 849 because it is only 1 degree further north and has similar diatom deposition intervals. Site 848 was the only site south of the equator, but was nearest to Site 849. We checked the records by displaying them all in Site U1338 equivalent depth and searching for ambiguous correlations.

### 3.2 The combined Site 849-Site U1338 age model

The Pacific Ocean lacks an orbitally resolving age model that covers the last 8 million years. We have constructed one by joining astrochronologically tuned stable isotope data from 0-3.6 Ma from Site 849 (Mix et al, 1995) to the tuned stable isotope data from 3.5 to 8.2 Ma from Site U1338 (Drury et al, 2016, 2018) Sites 849 and U1338 are the most central sediment records to the region we correlate, and provide good stratigraphic ties between the IODP Expedition 320/321 sites

and those cored by ODP Leg 138. See Supplemental Material section 3 for more details. The records are joined at 3.6 Ma because the revised Site 849 splice had gaps in the older part of the record (see supplemental material).

The isotope splice for Site 849 was assigned ages by correlation to the new Ceara Rise composite isotope stack in Wilkens et
5  al (2017) to provide age control between 0-5.3 Ma. For Site U1338, we used the astrochronology between 3.5 and 8.0 Ma developed by Drury et al. (2018) which revised the benthic stable isotope age model in Drury et al (2016). In this study, the better astro-magnetochronology developed at Site U1337 could be used to constrain ages at Site U1338 through the late Miocene (8.0-6.0 Ma) based on depth matching between U1337 and U1338, while the remaining 6.0-3.5 Ma interval was astronomically tuned using the benthic $\delta^{18}O$ data (Drury et al, 2017; Drury et al., 2018).

The 849 and U1338 age models were grafted together at 3.626 Ma to make the combined Site 849-U1338 age model, with Site 849 providing the younger section and Site U1338 providing the older one. The two isotope records match well at 3.6 Ma (Figure SM-3). Using the U1338 isotope stratigraphy in the early Pliocene also eliminates missing sections in the Site 849 record found at about 3.7 Ma and 4.2 Ma. The age model was propagated to all sites through use of the site-to-site
correlations derived from the depth matched physical properties and bulk composition profiles. See the supplemental material for more details and for the age model tables (Tables SM-14 to 17).

### 3.3 XRF scanning at Sites 849, U1335, U1337, and U1338

Only the XRF scanning data from Site 849 are new in this paper. XRF scanning for Sites U1335, U1337, and U1338 have been reported previously.. The major elements (Al, Si, K, Ca, Fe) and significant minor elements (Ti, Mn, Ba) were
measured using the Avaatech scanning XRF at the IODP Gulf Coast Repository. The element data was processed using the normalized, median-scaled (NMS) method of Lyle et al (2012. Raw and processed (but uncalibrated) data for Sites U1335 and U1337 are found in Shackford et al (2014). Similarly, raw and processed XRF scan data for Site U1338 are found in Lyle et al. (2012).

Much of the XRF data has been calibrated against discrete chemical analyses. Calibrated $CaCO_3$ % estimates from the U1338 XRF scan data are found in Lyle and Backman (2013) and use shipboard plus additional discrete $CaCO_3$ measurements. Other elements from the Sites U1337 and U1338 XRF scan data were calibrated with Inductively Coupled Plasma Mass Spectrometer (ICP-MS) analyses of bulk sediment for the elements Al, Fe, Mn, Ti, and Ba (Wilson, 2014).. $CaCO_3$ contents were calibrated to shipboard $CaCO_3$ analyses supplemented by additional $CaCO_3$ data found in Wilson
(2014). Discrete bio-$SiO_2$ analyses, where bio-$SiO_2$ was extracted using the KOH dissolution technique (Olivarez Lyle and Lyle, 2002), are reported in Wilson (2014) and used to calibrate the Site U1337 and U1338 XRF bio-$SiO_2$. The bio-$SiO_2$ percentages are reported without structural water—since there is ambiguity about the dependence of bio-$SiO_2$ structural water on the organisms that make the tests.

Clay was estimated by assuming that clay has constant $TiO_2$ and using the calibrated XRF $TiO_2$ estimate. Clay is assumed to have the same composition as Taylor and McLennan (1995) upper continental crust, which has 0.3% Ti. Clay-bound Si was estimated using the Taylor and McLennan (1995) Si/Ti ratio for upper continental crust. Bio-$SiO_2$ was calibrated to the total

5    Si minus the clay component (Lyle and Baldauf, 2015). See Supplemental Material Section 4.1 for more details. The calibrated XRF scan data for Site U1337 and for Site U1338 are listed in Supplemental Tables SM-19 and SM-20. Site U1335 XRF scanning $CaCO_3$ data are calibrated to shipboard $CaCO_3$ % and are found in Wilson (2014). Processed but uncalibrated Site U1335 XRF scanning data for other major and trace elements are found in Supplemental Table SM-18 along with the calibrated $CaCO_3$% estimate.

XRF scanning data for Site 849 are found in Supplemental Table SM-21 and are discussed in Supplemental Material Section 4.2. Shipboard $CaCO_3$ % measurements from Mayer et al (1992) were used to calibrate the Site 849 XRF scan $CaCO_3$ %, while discrete bio-$SiO_2$ analyses were used to calibrate the bio-$SiO_2$. The Bio-$SiO_2$ calibration data for Site 849 are found in Table SM-22. To estimate clay content we correlated the XRF $Fe_2O_3$ NMS to a $^{232}$Th estimate of clay from the upper part of

the sediment column (Winckler et al, 2008). We used the XRF $Fe_2O_3$ because Ti contents were often below the XRF detection limit deeper than 130 m CCSF (Composite Coring Depth below Sea Floor) and because the oxide bound Fe is minimal. We used barite data from Ma et al. (2015) to calibrate the Site 849 XRF Ba data.

The XRF scan data extend beyond 0-8.2 Ma for Sites U1335, U1337, and U1338, to basalt crust at Sites U1337 and U1338,

but only to a depth of 301.2 m CCSF at Site U1335. The records for Sites U1335 and U1337 cover the entire Neogene, while the base of Site U1338 is just below the early/middle Miocene boundary. Because of time constraints, Site 849 was scanned only to 7.2 Ma (286.72 m CCSF).

**3.4 $CaCO_3$ % estimates from GRA wet bulk density, Sites 848, 849, 850, and 851**

It has long been known that variations in bulk density in the eastern equatorial Pacific are correlated to changes in $CaCO_3$

content (Mayer, 1979; Mayer, 1991; Hagelberg et al., 1995; Reghellin et al, 2013). $CaCO_3$ can be calculated from wet bulk density in the equatorial Pacific because the sediments are essentially a mixture of bio-$SiO_2$ and biogenic $CaCO_3$ with less than 10% clays. $CaCO_3$-rich sediments have higher bulk density than porous bio-$SiO_2$-rich sediments (Mayer, 1991). After correcting for burial compaction that reduces porosity with depth and causes an increasing trend in bulk density, the variation in the "decompacted" bulk density can be converted to estimated $CaCO_3$ by correlating shipboard $CaCO_3$ data to

the decompacted GRA bulk density. Here we provide $CaCO_3$ estimates from GRA bulk density for Sites 848, 850, and 851 along the revised splices we made. We also estimated $CaCO_3$ at Site 849 from bulk density in addition to the XRF estimate in order to compare the XRF and bulk density estimates of $CaCO_3$. When compared with shipboard measurements the GRA-derived $CaCO_3$ % estimate (sd of difference from measured ±5.86%; see Figure SM-4 in the supplemental material) was

somewhat damped with respect to the discrete measurements of CaCO$_3$ % at Site 849. The XRF-derived CaCO$_3$ estimate (sd of difference from measured: ±3.86%) has a range that better matches that of the shipboard discrete measurements. Nevertheless, both estimates capture the same trends in the CaCO$_3$ profile. See Section 5 of the Supplemental Material for more details. Estimated CaCO$_3$% data are in Supplemental Tables SM-24 to SM-27.

## 3.5 Mass Accumulation Rate (MAR) calculation

The bulk sediment percentage data at each site were converted to MAR using the Site 849-U1338 age model, measured physical properties, and the bulk chemical data. The composition and physical properties data were interpolated to even 1-2 kyr spacing, then smoothed with a 20-point binomial filter before being resampled at 10-kyr intervals to accommodate probable errors in the age model and site-to-site correlations. MAR calculations are much more sensitive to age errors than age profiles of sediment components because they depend on the differential of depth versus time (sedimentation rate) as well as the time-depth correlation at each site. Bulk MAR was calculated by multiplying the dry bulk density by the sedimentation rate from the age-depth relationship. Individual component MARs were calculated by multiplying by the fraction of that component in the dry sample by the bulk MAR.

Bio-SiO$_2$, BaSO$_4$, and clay MAR records were created at the sites where both XRF data and independent discrete measurements for bio-SiO$_2$ and a clay-bound element existed. Sites U1337, U1338, and 849 had data suitable to calibrate the records, as discussed more in the Supplemental Material. Discrete chemical analyses have never been done for Site U1335, so only the CaCO$_3$ record is calibrated at this time, using shipboard CaCO$_3$% data from IODP Expedition 320/321. Ratios of uncalibrated Si to Ti from Site U1335 suggest that the sediments have low bio-SiO$_2$ contents. MAR data are found in Supplemental Tables SM-28 to SM-34

## 4 Eastern Equatorial Pacific Carbonate Deposition Patterns

We primarily describe the factors that have shaped CaCO$_3$ % and MAR profiles in the Pliocene and Pleistocene, because there are good examples of both production and dissolution dominated intervals since 5.3 Ma and because the complex nature of the LMBB interval merits additional work. CaCO$_3$ production has a latitudinal distribution and is highest at the equator (Honjo et al, 1995). Conversely, dissolution signals are strongest at deep sites underneath lower CaCO$_3$ production in latitudes away from the equator. Because the East Pacific Rise is roughly N-S in orientation, the depth effect imposes a longitudinal overprint upon the latitudinal production pattern. Furthermore, a 100-kyr cyclicity of CaCO$_3$ content first appears between 1.5 and 2 Ma and increases to the present, much earlier than the 100-kyr cycles of oxygen isotopes.

The CaCO$_3$ % and CaCO$_3$ MAR profiles at all 7 sites exhibit long period variability and orbital cyclicity (Figures 2 and 3). We have used the CaCO$_3$ % profiles to define five low long-period CaCO$_3$ % transients (Pliocene-Pleistocene Low CaCO$_3$

intervals, PPLCs) in Table 2. We use $CaCO_3$ % rather than MAR because (1) the percentage profiles can be directly measured as the sediment sections are processed at sea and they are the basis for most CCD reconstructions, (2) the $CaCO_3$ % variability is chronostratigraphically synchronous across the east central Pacific, (3) the results are easily compared to earlier observations from other cores and drill sites where only $CaCO_3$ % data are presented (e.g., Mayer et al, 1986, Farrell et al., 1991), and (4) the large changes of $CaCO_3$ produce the characteristic seismic horizons found in the eastern Pacific (Mayer et al, 1986; Bloomer and Mayer, 1997). We then use MARs and other elemental data to understand the causes of each PPLC.

### 4.1 Low $CaCO_3$ % Intervals in the Pliocene and Pleistocene

Two primary factors cause low $CaCO_3$ intervals—excess bio-$SiO_2$ production and/or higher than average relative $CaCO_3$ dissolution. Higher dissolution can result from an increase of dissolution at the sea floor, or a reduction of $CaCO_3$ surface production with little change of sea-floor dissolution. The late Miocene and early Pliocene have high $CaCO_3$ variability in the LMBB interval largely driven by variability in relative bio-$SiO_2$ and $CaCO_3$ production. Low $CaCO_3$ intervals are coherent through all 7 records and have lowest $CaCO_3$ in deeper sites away from the highest productivity. In Table 2 and Figure 2 we have defined five PPLCs. The same low carbonate intervals can also be found in the low-resolution profiles of Deep Sea Drilling Project (DSDP) Sites 572 and 573 from Farrell and Prell (1991), with some age shifts resulting from the older less accurate age model they used. .The lowest $CaCO_3$ % occurs at about 4 Ma (Figure 2, PPLC-4C, 3834-4093 ka), and since that time $CaCO_3$% has slowly climbed to peak at about 500 ka, just before the initiation of PPLC-1.

The ~100 kyr $CaCO_3$ variation (Hays et al, 1969; Farrell et al, 1989; Murray et al, 2000) characteristic of the Pleistocene equatorial Pacific dominates the $CaCO_3$% records after 1.9 Ma and the $CaCO_3$ variation is highest after 1 Ma (Fig. 2). The large $CaCO_3$ swings are most prominent in Sites U1337 and U1335, which are deep and relatively far from the equatorial productivity band. Sites along the Leg 138 transect have the smallest variability because they are also the shallowest sites. Earlier studies have found that $CaCO_3$ % is high in Pleistocene glacial climate intervals and is low in interglacials (Chuey et al., 1987; Farrell and Prell, 1989). Development of ~100-kyr cyclicity in the carbonate record predates the development of 100-kyr periodicity in the oxygen isotope record by about 1 Myr. The 100-kyr $CaCO_3$ cycles are discussed further in sections 4.6 and 5.2. The older part of each record in Fig. 2 varies at a slower pace than 100 kyr.

### 4.2 Potential sediment focusing at U1337

Except for Site U1337, Bulk MAR records are similar throughout the eastern equatorial Pacific, indicating that much of the variability in an individual record is caused by regional factors, not local sedimentation processes (Figure 3). Site U1337 has anomalous deposition that may be caused by local sediment focusing (Francois et al, 2004; Lyle et al, 2005a; Tominaga et al., 2011; Marcantonio et al, 2014). Variability of the bulk sediment MAR is typically caused by changes of $CaCO_3$ MAR that are coherent regionally. However, Site U1337 is unique because it clearly has much higher bio-$SiO_2$ and clay deposition

unrelated to $CaCO_3$ burial primarily between about 4.5 and 3 Ma, and also between 6.2 and 5.4 Ma. The two intervals may have had higher than average local deep current activity to sweep sediments to Site U1337. We propose that deposition at Site U1337 is the result of local sediment focusing adding fine sediments to the sediment column at the drill site (Marcantonio et al, 2014; Lyle et al, 2014; Lovely et al, 2017). When deep currents are moderate in strength, sediment
focusing preferentially moves the fine fraction of sediments (Lyle et al, 2014; Lovely et al, 2017) as is observed in the Site U1337 record. Our interpretation is supported by seismic profiles and bottom bathymetry, which find active erosional channels around the northern and eastern edges of the Site U1337 survey area (Figure SM-1b, SM-5, Supplemental Material Section 7; Pälike et al., 2010).

Surprisingly, the sediment focusing at Site U1337 does not strongly affect the $CaCO_3$ % profile, but does result in anomalously high sedimentation rates, lower average $CaCO_3$ contents, and higher than expected $CaCO_3$ MAR. For example, average $CaCO_3$ % for Site U1337 is 39.5% versus 54.5% for U1335, despite Site U1335 being 1.5° further north and further away from equatorial production. Similarly, $CaCO_3$ MAR at Site U1337 averages 20% higher than the $CaCO_3$ MAR at Site U1338 for the last 8 Myr despite being further from the equator than Site U1338 (Fig 3). Observations at Site U1337 suggest
that focusing over the Neogene enhances $CaCO_3$ MAR to some extent, but more greatly affects the MAR of finer sediment fractions.

### 4.3 Observed high production intervals: PPLC 5 (4465 to 4737 ka) and PPLC 2 (1685 to 2135 ka)

With a proper age model and with XRF scanning data it is possible to distinguish a $CaCO_3$ low caused by elevated bio-$SiO_2$ production from intervals caused by higher $CaCO_3$ dissolution at the sea floor. High bio-$SiO_2$ burial during Pliocene and
Pleistocene diatom deposition intervals can be distinguished by (1) low $CaCO_3$ content, presumably by dilution with diatom bio-$SiO_2$ for sites near the equator, (2) little change in clay MAR but large change in bio-$SiO_2$ content and bio-$SiO_2$ MAR, (3) increase or little change in $CaCO_3$ MAR, and (4) if there is a latitudinal transect, lowest $CaCO_3$ % and high diatom deposition will be found at sites closest to the equatorial upwelling zone. Sites away from the equator may actually have higher $CaCO_3$% and $CaCO_3$ MAR in the equivalent interval presumably because elevated nutrients stimulate higher
calcareous nannofossil production relative to time intervals on either side. The low $CaCO_3$ near the equator is caused mostly by dilution of $CaCO_3$ by high fluxes of bio-$SiO_2$ to the sea floor.

PPLC-5 (4737-4465 ka) is the last diatom deposition interval of the LMBB and exhibits all the characteristics of a high production interval. Sites 849 and 850, sites straddling the equator at 5 Ma (Table 1), exhibit a pronounced $CaCO_3$ % low at
this time (Figure 2), but sites farther away from the equator exhibit much smaller $CaCO_3$ % change (Sites 848, 851, and U1338) or even slight $CaCO_3$ % increases (Sites U1337, U1335). All sites have relatively high or unchanged $CaCO_3$ MAR within PPLC 5 (Figure 3). Bio-$SiO_2$ data is only available for Sites 849, U1338, and U1337, and the records exhibit a large

bio-SiO$_2$ MAR peak at Site 849 (Figure 4), a small peak at Site U1338 (Figure 6), and ambiguous change at Site U1337, exacerbated by potential sediment focusing there.

PPLC-2 (2135-1685 ka) is found at the equator at Site 849, but is also found at Site U1338. The equivalent of the PPLC-2 interval has also been reported to the east, near the Galapagos Islands (Sites 846 and 847, Farrell et al, 1995; Site 846, Lawrence et al., 2006; Site 1240, Povea et al., 2016). There are clear bio-SiO$_2$ MAR peaks at Site 849 (Figure 4) that match in time and magnitude with a similar interval from Site 1240 (Povea et al, 2016). Major bio-SiO$_2$ MAR highs within PPLC-2 align with glacial intervals in the oxygen isotope series between Marine Isotope Stage (MIS) 78 and MIS 60, even though the highest bio-SiO$_2$ MAR is in an interglacial (MIS 75). CaCO$_3$ % lows within PPLC-2 don't always align with high bio-SiO$_2$ MAR, indicating CaCO$_3$ dissolution has also occurred. At Site U1338, low CaCO$_3$% and highest bio-SiO$_2$ MAR are not aligned, evidence that a dissolution overprint exists.

The level of dissolution can be assessed by the CaCO$_3$:BaSO$_4$ ratio (Figure 5; Supplemental Table SM-22). Lyle and Baldauf (2015) proposed that the CaCO$_3$:BaSO$_4$ ratio is a good measure of relative preservation, since CaCO$_3$ preservation is highly variable, but Ba is better preserved, and the preservation is relatively constant (Dymond et al, 1992; Balakrishnan Nair et al, 2005; Griffith and Paytan, 2012). In addition, particulate Ba rain caught in sediment traps is proportional to the C$_{org}$ rain, so normalizing to Ba largely normalizes out the production variability and leaves a signal dominated by CaCO$_3$ dissolution. Both bio-Ba and CaCO$_3$ rain are well-correlated to particulate organic carbon rain (Dymond and Collier, 1996), so changes in the CaCO$_3$:BaSO$_4$ ratio primarily mark changes in the relative preservation of CaCO$_3$ (Lyle and Baldauf, 2015). Highest CaCO$_3$:BaSO$_4$ ratios in the XRF data set are found at Site 849, which is shallower and has about twice the sedimentation rate of Site U1338 or U1337 (Table 2). As expected, the ratio indicates lower dissolution while compensating for the higher production at Site 849.

PPLC-2 has generally low CaCO$_3$:BaSO$_4$ ratios which indicates that the interval not only has higher bio-SiO$_2$ burial indicative of high surface productivity (Fig 4) but also higher dissolution (Fig 5). Production and dissolution don't have to be mutually exclusive and can be superimposed to produce a low CaCO$_3$ interval.

**4.4 Early Pliocene extensive CaCO$_3$ dissolution: PPLC-4 (4093-2915 ka)**

Immediately after the LMBB ends there is the PPLC-4 interval of very low CaCO$_3$ %. The minimum is found throughout the eastern and central equatorial Pacific west of the East Pacific Rise. The base of PPLC-4 constitutes the acoustic impedance contrast that causes the Pliocene 'green' seismic horizon (Mayer et al, 1986; Bloomer et al, 1997). At Site 850, Bloomer and Mayer (1997) identified a series of seismic horizons that can be linked to the CaCO$_3$ variability associated with PPLC-3, 4, and 5.

PPLC-4 is made up of 3 low $CaCO_3$ % intervals between 2915 and 4093 ka (Table 2) with lower $CaCO_3$ % and more consistently low $CaCO_3$ % at sites further away from the equator. The lowest $CaCO_3$ % in all the records is found at U1337 between 4500 and 3500 ka because apparent sediment focusing has caused extensive clay and bio-$SiO_2$ dilution in addition to $CaCO_3$ % dissolution (Figures 2 and 3). Nevertheless, relatively shallow sites along the Leg 138 transect also exhibit coherent $CaCO_3$ change, as exhibited by the standardized $CaCO_3$ % records (Figure 2b). The low bulk MAR and $CaCO_3$ MAR over the PPLC-4 interval indicates that dissolution was a major factor shaping the $CaCO_3$ record. The $CaCO_3$:$BaSO_4$ records (Fig 5) also clearly show the low $CaCO_3$:$BaSO_4$ ratios indicative of dissolution in the PPLC-4 intervals. They have the lowest $CaCO_3$:$BaSO_4$ ratios in the time series and the lowest $CaCO_3$ MAR.

The PPLC-4 set of dissolution-induced $CaCO_3$ % lows occurs at an important juxtaposition of tectonic and environmental influences. Slow closure of the Central American Seaway was severing connections between the Caribbean and Pacific. However, extensive changes were also going on in global climate, so it is currently impossible to isolate the seaway closure effect from other changes in the eastern Pacific (Molnar, 2008; O'Dea et al, 2016; Molnar, 2017).

### 4.5 PPLC-3 (2684-2248 ka): $CaCO_3$ dissolution at the initiation of Northern Hemisphere glaciation

PPLC-3 consists of two dissolution intervals separated by a distinctive triplet of high $CaCO_3$% peaks marked on figures 2 and 5. PPLC-3 can best be seen in the stacked and standardized $CaCO_3$ % profile (Figure 2, bottom). The PPLC-3 dissolution intervals surrounding a high carbonate triplet are also found at DSDP Sites 573/574 (Farrell and Prell, 1991). The consistent expression of the PPLC-3 $CaCO_3$ % lows merited their inclusion in this list. Deeper sites within the eastern Pacific have prominent lows around the triplet of high $CaCO_3$ %, while records from the shallower Leg 138 sites have weaker variability. Along the relatively shallow Leg 138 transect (~3800 m water depth) PPLC-3 lows are most pronounced south of the equator at Site 848 (3°S) and north of the equator at Site 851 (2.7°N), as expected if dissolution is superimposed upon an equatorial maximum of $CaCO_3$ production. The timing of PPLC-3 suggests an association with the 2.7 Ma initiation of northern hemisphere glaciation (Haug et al, 2005), or at least with changes in abyssal circulation at that time. Dissolution at PPLC-3 can also be determined by the $CaCO_3$:$BaSO_4$ ratio (Figure 5). The $CaCO_3$:$BaSO_4$ ratio around PPLC-3 clearly shows similar responses in all the records we have, with two low $CaCO_3$ intervals surrounding the $CaCO_3$ % high.

### 4.6 PPLC-1 (51-402 ka): low $CaCO_3$ interval among the 100-kyr Pleistocene cycles

The PPLC-1 interval is superimposed upon the classic Pleistocene Pacific 100-kyr glacial-interglacial carbonate cycles (Figure 5). Significant 100-kyr $CaCO_3$ cyclicity first appears at about 1900 ka in the eastern Pacific, well before appearance of 100-kyr benthic oxygen isotope cycles at ~900 ka that characterize the end of the Mid-Pleistocene Transition. If Sites U1338 and 849 are compared it is clear that dissolution is a major factor in the cycles. Site 849 is the shallowest site and has much higher $CaCO_3$:$BaSO_4$ (less dissolution) than the other sites plus much weaker 100 kyr cycles. Dissolution apparently increases the amplitude of the $CaCO_3$ cycles within the Site U1337 and Site U1338 record. Not only does dissolution

increase the 100-kyr cycle amplitude but also selectively removes higher frequency variability in the $CaCO_3$:$BaSO_4$ ratio that are apparent at Site 849. Highest $CaCO_3$ preservation as well as highest $CaCO_3$ % occurs just before the beginning of PPLC-1. The time offset of the low $CaCO_3$:$BaSO_4$ ratio at Site 849 results from relatively weak dissolution in the early part of PPLC-1. These cycles largely match the Pleistocene CCD changes found by Farrell and Prell (1989) at around 140°W.

## 5 Discussion

Two major processes affect carbonate burial in the equatorial Pacific. High production within the surface waters of the equatorial productivity belt may increase $CaCO_3$ rain but may also dilute $CaCO_3$ sediment concentrations with bio-$SiO_2$. Dissolution by deep waters from the North Pacific rich in dissolved inorganic carbon (DIC) and low in $[CO3]^=$ may reduce $CaCO_3$ burial by increased $CaCO_3$ dissolution at the sea floor. As noted in the introduction, Antarctic abyssal waters flow into the North Pacific basin to the west of Hawaii, and the return flow travels south from the northeast Pacific along the west flank of the East Pacific Rise (Johnson and Toole, 1993; Hautala, 2018) across the drillsites studied in this paper. All the east Pacific sites have been exposed to the same modified bottom water streaming from the north.

$CaCO_3$ burial is the net difference between biogenic $CaCO_3$ production and dissolution at the sea floor. If $CaCO_3$ production decreases but dissolution at the bottom stays relatively constant, lower $CaCO_3$ burial will also occur. Dissolution effects can be distinguished from those caused by productivity, in particular, dilution by high bio-$SiO_2$ deposition, by using burial fluxes (MAR) along with other sediment chemical data. $BaSO_4$ MAR are especially useful to distinguish between dissolution and reduced production as the primary cause of an interval of low $CaCO_3$ %.

The CCD concept is most useful at a reconnaissance scale. But, using a ratio such as $CaCO_3$:$BaSO_4$ to normalize for different $CaCO_3$ rain rates provides the best indicator of relative dissolution at any given drill site (Figure 5). The ratio of benthic to planktic foraminifera could also be used to assess local carbonate dissolution, but requires measurements in addition to the XRF scans.

We find that dissolution is most pronounced in the early Pliocene (PPLC-4) and is evidence that a major reorganization of carbon storage in the Pacific occurred then. We also found that bio-$SiO_2$ dilution shapes the record, especially in PPLC-5 and PPLC-2. More XRF profiles, especially from ODP Leg 138 and ODP Leg 202 would be very useful to better understand the Pacific carbon cycle and the regional to global biogenic sedimentation patterns that are observed.

**5.1 Disentangling production versus dissolution: Eastern Tropical Pacific Sedimentation since the Miocene Climate Optimum (14 Ma)**

Profound changes in equatorial Pacific biogenic deposition from the middle Miocene to Recent were recorded at Site U1338 (Lyle and Baldauf, 2015). Dilution of $CaCO_3$ by bio-$SiO_2$ deposition appears to be relatively common in the middle and late
Miocene (Holbourn et al, 2014; Lyle and Baldauf, 2015). For the most part, $CaCO_3$ burial was high since the end of the Miocene Climate Optimum, except within the late Miocene carbonate crash, 11-8 Ma (Lyle et al., 1995; Roth et al., 2000; Lyle and Baldauf, 2015). Poorest $CaCO_3$ preservation occurred at about 9.7 Ma. The LMBB, from 8 to about 4.5 Ma, occurred in the equatorial region west of the East Pacific Rise after the carbonate crash (Farrell et al, 1995; Lyle and Baldauf, 2015). Consistently high sedimentation rates and biogenic deposition characterizes the LMBB with considerable
variations in bio-$SiO_2$ and $CaCO_3$ burial. The last diatom production interval within the LMBB (PPLC-5) ended at 4465 ka, within the early Pliocene (see Section 5.2.1 for further discussion of the LMBB). In the LMBB, the most diatom-rich intervals are a mixture of mat forming and upwelling diatom species, but the upwelling species extended over much longer time ranges than the high diatom deposition intervals and are not restricted to them. Northern subtropical diatoms were abundant at the beginning of the LMBB, and were replaced by diatoms now associated with the southern hemisphere
subtropics by 4.4 Ma (Lyle and Baldauf, 2015).

Changes in biogenic deposition apparently result from variable responses under different nutrient/upwelling conditions by different plankton. At Site U1338, $BaSO_4$, bio-$SiO_2$, and $CaCO_3$ MAR reveal a major drop in burial at the end of the LMBB but also some second order cyclicity (Fig 6). PPLC-4b and -4C intervals are associated with low biogenic deposition of bio-
$SiO_2$ and bio-Ba, so $CaCO_3$ dissolution within those intervals may have been enhanced by low relative production of $CaCO_3$. Estimated clay deposition varied much less than that of the biogenic components, especially across the LMBB. Since Ba production is linear with respect to organic carbon export production (Dymond et al, 1992), the $BaSO_4$/clay ratio should be relatively constant if changes in dust-delivered Fe were the primary cause of the large-scale changes in productivity. At the end of the LMBB, the $BaSO_4$/clay ratio dropped (top graph on Figure 6) but the clay MAR remained relatively constant
(bottom graph on Figure 6), indicating that changes in dust delivery of Fe was not the main cause to end the LMBB. In contrast, in PPLC-2, the BaSO4/clay ratio was essentially constant despite relatively high $BaSO_4$ MAR, indicating that changes in dust-derived Fe may have been critical to the change in surface productivity.

The observed changes in apparent productivity exceed the envelope encompassed by seasonal change in the JGOFS Pacific
sediment trap experiment (Honjo et al, 1995; Dymond and Collier, 1996). Further work to understand the biogeochemical response to long-term climate change should be done. Not only were there long intervals of high biogenic deposition, but also the ratios of the different biogenic hard parts varied by large amounts and different species within the phytoplankton participated at different times.

**5.2 Primary Productivity in the eastern equatorial Pacific (PPLC-5 & -2)**

The LMBB (with PPLC-5) and PPLC-2 at 2 Ma result from major changes in primary productivity in the eastern Pacific. Apparently conditions conducive to high primary production were in place for hundreds of thousands of years (Table 2) modulated by Milankovitch cycles. Here we discuss the most likely causes of these prolonged phases.

**5.2.1 The Late Miocene Biogenic Bloom and PPLC-5**

The LMBB has multiple intervals of high biogenic deposition after 8 Ma (Figure 3; Farrell et al, 1995; Lyle and Baldauf, 2015), ending with PPLC-5. The significant difference in biogenic sediment accumulation associated with the LMBB is easily observed by the change in slope of the age-depth curves at about 4400 ka (Fig 7). Each drill site exhibits slower sedimentation rates after ~4400 ka, marking the end of the LMBB. In Figure 1, the spatial pattern of the LMBB is also

shown by a ratio of the thickness of the older sediments (8-4.5 Ma) from the LMBB to that of the younger ones since the end of the LMBB (4.5-0 Ma), referred to on Figure 1 as the LMBB/Pliocene ratio. Because only two time horizons need to be identified, and because this metric uses only sediment thickness, other sites can be added easily to the map using the published site descriptions.  Additional Leg 138 Sites (Sites 844, 845, 846, 847, 852, 853, and 854) plus sites from ODP Leg 202 (Sites 1238, 1239, and 1241) and DSDP Sites (83, 158, and 572) were used to map the regional extent of the LMBB and

its strength.

Despite additional sediment compaction and the shorter total time span within the older interval (3.5 versus 4.5 Myr), the thickness of sediment section deposited during the 8-4.5 Ma time interval from many eastern Pacific drill sites is often more than that deposited in the 4.5-0 Ma interval. Since the sediments are primarily biogenic in origin, high ratios reflect high

productivity in the LMBB. During the LMBB the largest ratios are found in the region from roughly 88°W to 120°W, just north of the modern equator (orange shading in Figure 1). Factoring in Pacific plate motion, these sites were at the equator around 6-7 Ma.  East of the Galapagos, sites have  higher biogenic sedimentation after the LMBB, so the LMBB has a regional expression.

The cause of the variability during the LMBB needs further investigation and documentation. Many regions worldwide have intervals of high production in the late Miocene and early Pliocene, which may have a common cause. Globally, high biogenic deposition in the Indian Ocean, North and Southwest Pacific, and the Benguela upwelling zone define a proposed global late Miocene production interval (Farrell et al, 1995; Filippelli, 1997; Dickens and Owen, 1999; Diester-Haass et al., 2002; Grant and Dickens, 2002).

High biogenic burial is common within the late Miocene, but the timing varies significantly between regions. Dickens and Owen (1999) propose that upwelling in the Indian Ocean was high between 9 and 3.5 Ma but most intense between 6 and 5

Ma. Dickens and Barron (1997) recognized an interval of high pennate diatom deposition between 5.9 and 5.0 Ma in the subarctic North Pacific and noted that the timing matched pennate diatom layer deposition in the equatorial Pacific (Kemp and Baldauf, 1993). Highest biogenic deposition in the Benguela current region also occurred in the period between 7 and 4.5 Ma (Diester-Haass et al, 2002). Within the Expedition 320/321 sites and the ODP Leg 138 sites along the 110°W transect, there is also high but variable biogenic MAR between ~8 Ma and the end of PPLC-5 at 4.47 Ma that indicate high primary production. The highest biogenic deposition in the eastern equatorial Pacific was between 7 and 6.5 Ma. Broadly, there is high biogenic production and burial in the late Miocene after 8 Ma.

Hypotheses to the cause of high primary productivity vary. Filipelli (1997) proposed that the uplift of the Himalayas (Molnar et al., 1993) and late Miocene intensification of the Asian Monsoon caused higher weathering and larger transport of nutrients to the oceans. Diester-Haass et al (2002) suggested that reorganization of ocean circulation is likely to have played an important role, as well as intensification of trade winds. Recent reconstructions of Himalayan uplift and development of the Asian monsoon have found that much of the uplift occurred prior to the late Miocene, weakening the weathering hypothesis (Tada et al, 2016).

Higher biogenic deposition within the equatorial Pacific broadly occurred within the late Miocene but also extends through the Pliocene in the easternmost equatorial Pacific (Fig 8). Million-year increments of $CaCO_3$ MAR and noncarbonate (mostly bio-$SiO_2$) MAR are shown from drill sites in the equatorial zone from near the 81°W longitude of the coast of Ecuador to almost 120°W, halfway to Hawaii. For all but the easternmost part of the equatorial transect high $CaCO_3$ MAR is confined within the time frame of the LMBB. However, east of about 100°W on the Nazca Plate, the bio-$SiO_2$ deposition is much higher in the Pliocene and Pleistocene, as shown by the increased noncarbonate (mostly bio-$SiO_2$) MAR post-LMBB.

A similar lack of a common pattern is found worldwide. Highest deposition in the eastern equatorial Pacific LMBB is in the region near the equator between about 88°W and 120°W (Fig 1), and the highest MAR was at the beginning of the LMBB, around 7 Ma. Site 1085 within the Benguela Current has highest biogenic MAR at the end of the LMBB interval (Diester-Haass etal, 2002). Peak diatom deposition in the subarctic North Pacific occurred in the middle of the LMBB interval, between 6 and 5 Ma (Dickens and Owen, 1999; Dickens and Barron, 1997). And, in the California current region, diatom deposition decreased offshore at the beginning of the LMBB (7.5 Ma) to become more confined to the coastal region, implying decreased overall bio-$SiO_2$ deposition after 7.5 Ma (Barron et al., 2002).

Farrell et al (1995) described the abrupt drop in biogenic deposition along the 110°W transect at the end of the LMBB but continued high deposition to the east. They proposed that the change was related to movement of biogenic depocenters resulting from Central American Seaway (CAS) closure. Sites drilled on the Carnegie Ridge by ODP Leg 202 support a major increase in biogenic sedimentation in the east associated with CAS closure (Fig 8). CAS closure, being a tectonic

process, took place over a long period of time (O'Dea et al, 2016). Initial shoaling of sea passages to around 1000 m depth (middle bathyal), occurred between 13 and 12 Ma (Duque-Caro, 1990), while the Caribbean and eastern Pacific stable isotopes became distinct after 4.2 Ma (Haug et al., 2001). The major terrestrial North and South American terrestrial mammal exchange at about 2.6 Ma is generally used to mark CAS closure (O'Dea et al, 2016).

Both Sites 1238 and 1239, near Ecuador, are situated on crust formed in the late early Miocene. Both essentially have a hiatus until ~8 Ma (ODP Leg 202 Shipboard Scientists, 2003). Biogenic sedimentation at both Sites 1238 and 1239 rapidly increased after 7 Ma, peaked between 3 and 5 Ma, and continued to remain high up to the Holocene. Apparently, the easternmost Pacific became more productive because of the formation of the Isthmus of Panama. Nevertheless, high levels

of paleoproductivity found after 8 Ma worldwide suggest that regional production intervals are superimposed upon a global signal. Now that high-resolution records of biogenic deposition are available from the east-central Pacific, there is a need to revisit the timing of regional changes in biogenic deposition through the late Miocene globally.

### 5.2.2 Unravelling the driving force of equatorial productivity

When the LMBB and PPLC-2 are examined in detail, it is apparent that the dynamics of nutrient delivery to the equatorial Pacific cannot be ignored. Figure 4 shows the PPLC-2 interval from Site 849, and is comparable to Site 1240 in the Panama Basin (Povea et al, 2016) and Site 846 (Lawrence et al, 2006). There is clearly an orbital periodicity at Site 849, as well as depositional events on the <1 kyr resolution of the XRF record.

Similarly, within the LMBB at the XRF-scanned intervals of Site U1338 and Site 849 there is high bio-SiO$_2$ variability on the millennial scale based upon the XRF estimated bio-SiO$_2$:clay ratio, providing evidence that long bio-SiO$_2$ deposition intervals are made up of large numbers of individual bio-SiO$_2$ deposition events. These deposition intervals correlate to the intervals where diatom mats have been found in the equatorial Pacific cores, around 4.2 Ma, 5.5 Ma, and 7.3 Ma (Figure F23, Pälike et al, 2010; Kemp and Baldauf, 1993). As mentioned previously, clay deposition within the LMBB is about the

same as that through the Pliocene and Pleistocene (Fig 7), so the elevated biogenic MARs do not result from higher dust fertilization but instead apparently result from interoceanic reorganization of nutrient inventories (Ziegler et al, 2008). The BaSO$_4$/clay ratio (top of Fig 6) illustrates the higher relative deposition of biogenic sediments to clay within the LMBB, evidence for nutrient reorganization.

High productivity intervals in the eastern Pacific result from a combination of factors. There appears to be elevated production globally within the LMBB, for example, indicating higher availability of nutrients in surface waters generally in the late Miocene. However, timing of intervals of high productivity depends on how the nutrients flow to different upwelling regions, and provide insight into how the sub-surface oceans have changed. The appearance of PPLC-2 and its strong

expression to the east of our study (Lawrence et al, 2006; Povea et al, 2016), show that long intervals where the euphotic zone had better access to nutrients occurred well into the Pleistocene.

### 5.3 Monitoring changes in depth dependent dissolution: eastern equatorial Pacific CCD versus CaCO₃:BaSO₄

The CCD is a common metric for variations in the carbon cycle over time. Unfortunately, changes in the CCD can be difficult to interpret because changes in $CaCO_3$ production affect the CCD just as strongly as variations of abyssal dissolution. Early discussions of the Pacific CCD, for example, found that the depth to complete $CaCO_3$ dissolution was deeper under regions of higher $CaCO_3$ production at the equator (Berger, 1973; van Andel and Moore, 1974). Without regional drill sites that span the CCD within the same production regime it remains difficult to define CCD changes except in

low resolution or for abrupt large CCD movements like at the Eocene-Oligocene boundary (Coxall et al, 2005; Pälike et al, 2012).

### 5.3.1 CaCO₃ MAR to estimate changes in CCD

Lyle et al (2005b) used the gradient in $CaCO_3$ MAR with depth between drill sites to define changes in the Eocene CCD, where site depth is corrected for crustal cooling and sediment loading. Working backwards from $CaCO_3$ burial (MAR) has

many ambiguities, however, and local sedimentation anomalies at the drill sites used cause significant noise. For large changes as in the Eocene, the $CaCO_3$ MAR gradient approach works, but the weaker signal in the Pliocene and Pleistocene is hard to distinguish from the noise.

$CaCO_3$ MAR at any given location and time can be represented as a combination of the net focusing, rain, and dissolution as

in equation (1):

$$CaCO_3 \ MAR_T = F_T (CaCO_3 \ rain_T) - D_T \tag{1}$$

where the T subscript refers to time T, rain refers to the particulate $CaCO_3$ arriving at the bottom from surface $CaCO_3$

production, F is the sediment focusing factor, and D is the $CaCO_3$ dissolution at the sea floor. The variation of D responds to the chemistry of the regional water mass and depth of the sea floor; but regional variability is typically a depth function because of the large spatial scale of deep water masses. However, sediment focusing is locally variable and $CaCO_3$ production typically has a smaller spatial scale than the scale of water masses that control dissolution. Production and sediment focusing are thus difficult to control for in order to solve for the dissolution component.

An ideal set of sites to isolate the dissolution component of the carbonate signal would have similar productivity but would be at different water depths. Assuming production and focusing have remained the same at both sites over time, the depth

dependent dissolution can be isolated. By dividing the change in $CaCO_3$ MAR between the two sites by the difference in water depth between them in the past, corrected for depth change caused by crustal cooling and sedimentation (Lyle, 1997; also see Supplemental Material Section 7) produces the drop in $CaCO_3$ MAR per meter of water depth. Dividing the $CaCO_3$ MAR at the deeper site by this gradient of $CaCO_3$ MAR with water depth produces the additional depth needed to reduce the

$CaCO_3$ MAR to zero, marking the CCD. This was the approach of Lyle et al (2005) to determine changes in the Eocene CCD.

Two of the sites in this study (Site 851 and Site U1338) are now at roughly the same latitude throughout the last 5 million years and have similar modern particulate organic carbon standing stock from satellite estimates (58 mg/m$^3$ POC at Site 851;

57 mg/m$^3$ POC at Site U1338, NASA Goddard Space Flight Center, 2014). Site 851 is 446 m shallower than Site U1338, Hypothetically, if relative sediment focusing at both sites remained constant and productivity changed coherently, the gradient in $CaCO_3$ MAR from the deep site should extrapolate to zero $CaCO_3$ MAR at the CCD. Relative sediment focusing over long time frames in the pelagic regime appears reasonably constant (Liao and Lyle, 2014; Mitchell and Huthnance, 2013), and current-related anomalies can be discerned from the seismic profiles from site surveys. Sites chosen for drilling

were chosen for the lack of anomalous seismic character in both ODP Leg 138 and IODP Exp 320/321. The $CaCO_3$ MAR gradient using Sites 851 and U1338 is noisy but general trends can be discerned (Fig 3). The trends are consistent with trends in $CaCO_3$ MAR and $CaCO_3$:$BaSO_4$ as indicators of enhanced dissolution.

The CCD estimate using $CaCO_3$ MAR appears noisy over longer time frames because of a combination of relatively high

frequency changes (e.g., the 100-kyr cycles in the last million years) but also suffers from the noise resulting from building the trend with records from only 2 sites. Minor errors in correlation plus local changes in $CaCO_3$ production or burial are magnified in the CCD estimate. For example, Site U1338 occasionally had higher $CaCO_3$ MAR than Site 851, making for a negative CCD gradient with depth. Negative trends result from either $CaCO_3$ production anomalies relative to modern conditions or changes in sediment focusing. These intervals were removed from the CCD estimate (e.g., the gap in the record

between 2000 and 2250 ka; Table SM-35).

Figure 3 also displays the Pälike et al (2012) CCD change over this period, which was reconstructed at 250 kyr intervals. The average CCD depth is similar between Pälike et al (2012) and the Site 851-Site U1338 MAR CCD extrapolation, but the Pälike et al (2012) version has much smaller deviation of the CCD over time. Pälike et al (2012) use a graphical technique to

estimate the CCD where $CaCO_3$ MAR is plotted versus paleodepth of each drillsite. Site U1334 constrained the change of the CCD in the youngest part of the time interval. Ideally the sites used to constrain the CCD should be within a common depositional regime. Sites U1338 and Site 851 are within the equatorial regime at the time of the LMBB, but Site U1334, without $CaCO_3$ at the time of LMBB deposition, was located at 6°N, and in a region of much lower $CaCO_3$ production in the surface waters. The low $CaCO_3$ production at this distance from the equator causes the local CCD to be much shallower than

that at a site nearer high productivity (Berger, 1973). There are no drill sites in the equatorial region that pass through paleodepths of >5000 m in the late Miocene.

The unsmoothed CCD records by the MAR method are very noisy, so smoothing is necessary. Two levels of smoothing are used to make the CCD in Fig 3, 50 kyr and 750 kyr. The shorter time frame is adequate to show higher resolution changes, while the latter captures the longer trends. It is clear from these records that the late Miocene-Recent CCD has been strongly affected by changes in production. Since 8 Ma The LMBB interval has the deepest CCD as well as the highest $CaCO_3$ MAR at both Sites U1338 and 851. The relative $CaCO_3$ MAR between Site 851 and Site U1338 is especially noisy in the LMBB, with frequent times when $CaCO_3$ MAR was higher at the deeper Site U1338. Nevertheless, it is likely that higher $CaCO_3$ production over Sites 851 and U1338 drove the eastern Pacific CCD deeper during this interval, even though we suspect the magnitude of CCD deepening.

The CCD starts to shallow beginning at ~4800 ka, as the LMBB was fading. The shallowest CCD determinations do not precisely align with the PPLC-4 intervals although PPLC-4 is contained within a broad CCD minimum, indicating both a response to lowered $CaCO_3$ deposition and, as indicated by $CaCO_3:BaSO_4$, to increased dissolution. During PPLC-1, the low $CaCO_3$% is clearly averaged over broad variability in $CaCO_3$ dissolution. Nevertheless, low $CaCO_3:BaSO_4$ during the PPLC-1, 3 and 4 intervals also indicate that enhanced dissolution was an important factor in the low average $CaCO_3$% (Fig 5).

### 5.3.2 $CaCO_3:BaSO_4$ ratio to define $CaCO_3$ dissolution

Section 4.3 discusses how the $CaCO_3:BaSO_4$ ratio can be used as a site specific indicator of dissolution. We found that there is a common signal to the $CaCO_3:BaSO_4$ variability at the 4 sites with XRF scanning profiles (Fig 5). Standardizing the data to units of standard deviation from the site mean better illustrates the common signal among all the sites where Ba data are available (Fig 5b). The PPLC-3 and PPLC-4 intervals clearly have lower average $CaCO_3:BaSO_4$ at Sites U1335, U1337, U1338, and 849. These sites span from the equator to more than 5°N and are up to 1900 km apart. The common dissolution record thus covers a significant portion of the east-central Pacific. Lowest $CaCO_3:BaSO_4$ ratios are found in PPLC-4c and -4a, also shown by lower $CaCO_3$ MAR than in other intervals (Fig 3). $BaSO_4$ MAR during PPLC-4c and -4a are relatively high, so dissolution must have outpaced somewhat higher $CaCO_3$ production relative to stratigraphic intervals on either side (See Supplemental Tables SM-17 to SM-21). The difference between the CCD change and the $CaCO_3:BaSO_4$ probably results from the added $CaCO_3$ rain within PPLC-4. Similarly PPLC-2 has relatively high $CaCO_3$ dissolution, but not enough to make a $CaCO_3$ MAR minimum during the interval. The LMBB exhibits major intervals of high relative $CaCO_3$ deposition, although significant dissolution intervals appear at ~5810 and ~5350 ka and prior to 7500 ka (Fig 5). The high $CaCO_3:BaSO_4$ ratios in the LMBB broadly match the deep CCD in Figure 3.

The dissolution anomalies cover a large regional extent as expected from the large regional extent of deep water masses. PPLC-3a and -3b are very consistent across the set of drill sites, and mark dissolution on either side of a distinctive $CaCO_3:BaSO_4$ maximum all occurring as the northern hemisphere glaciations began.

### 5.4 Early beginnings of Pleistocene 100-kyr $CaCO_3$ cyclicity

5   Both the $CaCO_3$ % and the $CaCO_3:BaSO_4$ records illustrate the development of 100-kyr deep Pacific dissolution cyclicity (Figures 2 and 5). Surprisingly, the 100-kyr dissolution cyclicity began significantly before they strengthened towards the end of the mid-Pleistocene transition at ~900 ka, at which time 100-kyr benthic oxygen isotope cycles became prominent. The dissolution cycles isolated by $CaCO_3:BaSO_4$ begin at about 1900 ka (Fig 9a) and build in amplitude until MIS 16 at 655 ka. High $CaCO_3$ preservation is associated with periods of higher ice volume since about 1700 ka (MIS 58).

The appearance of 100-kyr power can also be tracked via wavelet time series of the stacked $CaCO_3:BaSO_4$ record (Fig 9b). Strong power is found at 100 kyr from 1900 ka going forward in time and in the brief interval between PPLC-3a and -3b. Relatively strong 41-kyr power ka is also apparent associated with the 100-kyr cycles since 1900 ka, and strong 41-kyr power is found at the end of the LMBB.

The early appearance of 100-kyr $CaCO_3$ dissolution cycles in the equatorial Pacific is interesting but problematic. There is a tendency even within the $\delta^{18}O$ 41kyr world for high $CaCO_3$ preservation to be associated with periods of heavier than average oxygen isotope cycles, and presumably cooler high latitude regions. Figure 9a compares the $CaCO_3:BaSO_4$ combined stack, where a high value indicates better $CaCO_3$ preservation, to the combined Site 849-Site U1338 isotope

record.

When both oxygen isotopes and $CaCO_3:BaSO_4$ have strong 100-kyr cycles (1000-0 ka), it is very clear that high $CaCO_3$ preservation is associated with glacial heavy isotope intervals. In the interval from 1000-1650 ka (MIS 25-59), that correlation is still strong when $CaCO_3:BaSO_4$ is compared to the smoothed oxygen isotope record. The correlation is weaker

before 1600 ka, but between 2400 and 2600 ka the high $CaCO_3$ preservation interval between PPLC-3a and -3b is also characterized by a heavy oxygen isotope signal.

The 100-kyr $CaCO_3$ cycles that appear in the Pacific could also appear in the deep Atlantic if there is cyclic weathering or variability of NADW formation. Unfortunately, there are few $CaCO_3$ records for the Atlantic Ocean that cover the period

between 1 and 2 Ma. Ruddiman et al (1989) have a $CaCO_3$ profile from Site 607, at a depth of 3426 mbsl. Because it is shallow relative to Antarctic Bottom Water (AABW) in the Atlantic, it is unlikely to be affected by dissolution and instead appears to be affected by changes in clay deposition over glacial-interglacial cycles (Broecker and Turekian, 1971; Bacon, 1984). The $CaCO_3$ % record at Site 607 changes from 41-kyr cyclicity to 100-kyr cyclicity at ~900 kyr, like the $\delta^{18}O$ record.

Harris et al (1997) developed a CaCO$_3$ dissolution index for the Ceara Rise and found that dissolution affected the CaCO$_3$ records below a depth of 4356 mbsl. The index compared CaCO$_3$ MARs at the deep Site 929 to the shallow Site 925, and found coherence with the oxygen isotope record of ice volume for the last million years. However, 100-kyr coherence was not very strong in the 1-2 Ma period. Since the dissolution record at the Ceara Rise primarily reflects oscillations in the penetration of AABW versus NADW around Ceara Rise, changes in NADW flow do not cause the Pacific oscillation. The differences between the Atlantic and Pacific CaCO$_3$ records imply the development of oscillations in deep storage of DIC in the Pacific, but little communication of this oscillation between ocean basins before the 900 ka mid-Pleistocene climate transition. The lack of strong cycles at the Ceara Rise may actually be caused by relatively strong NADW formation in the Atlantic prior to 1 Ma and but weakening NADW production and the development of NADW cyclicity only after the mid-Pleistocene climate transition (Kleiven et al., 2003).

For the 2.6 million years of the Pleistocene we observe an important linkage between CaCO$_3$ preservation and higher ice volume. Farrell and Prell (1989) have argued for a change in carbonate ion concentration variability of about 6% to drive the glacial-interglacial cycles since 800 ka. Hodell et al. (2001) have argued that the Indo-Pacific association of high CaCO$_3$ burial with glacials is evidence for cycles driven by shelf-basin fractionation of carbonate burial. During high sea level stands, the additional burial of CaCO$_3$ on shallow tropical shelves reduces deep CaCO$_3$ burial, while exposure of these shelves during glacials transfers CaCO$_3$ burial to the deep ocean and also consumes atmospheric CO$_2$ through carbonate weathering. The correlation that we observe between higher CaCO$_3$:BaSO$_4$ and heavy oxygen isotopes is supportive of a primary control of deep CaCO$_3$ burial by shelf-basin transfer through the Pleistocene, with ocean circulation transferring the signal mostly to the Pacific.

### 5.5 Origins of PPLC-4 and the Pliocene temperature maximum

Pleistocene changes in dissolution are adequately explained by shelf-basin fractionation, but the Pliocene PPLC-4 dissolution series requires additional factors. Most of PPLC-4 is associated with lower frequency dissolution variability prior to 2600 ka (Figs 5 and 9) and the PPLC-4 intervals are also marked by extreme lows in CaCO$_3$ MAR (Fig 3).

PPLC-4 occurs during the final closure of the CAS (O'Dea et al, 2016), which might link closure and the extreme dissolution. Bell et al (2015) suggest that NADW formation was strong prior to CAS closure and largely unaffected by changes in CAS geometry. Furthermore, they propose that the circulation effects of closure developed prior to 4 Ma. It does not appear that PPLC-4 dissolution is linked to flushing the Atlantic with NADW.

Poore et al (2006) have proposed that the deep sill at the Greenland-Scotia Ridge between 6000 and 2000 ka caused a radical increase in volume of Northern Component deep water production, based upon strong $\delta^{13}$C gradients between the North Atlantic-Southern Ocean-Pacific. Such an increase should increase basin-basin fractionation of CaCO$_3$ burial between the

Atlantic and Pacific. However, PPLC-4 also occurs immediately after the end of the LMBB, marking a major drop in biogenic $CaCO_3$ rain regionally in the Pacific, and is also a period of high estimated atmospheric $CO_2$ (Seki et al, 2010; Stap et al, 2016). PPLC-4 occurs during an interval of warm tropical SST after the SST minimum at 5.4 Ma (Herbert et al., 2016). In other words, many important changes in the global biogeochemical cycles and global climate occurred at the time of

PPLC-4, and it is as yet unclear how much these changes are related to CAS closure.

Nevertheless, it is possible to say that during the early Pliocene the Atlantic was preferentially flushed of DIC, making positive $\delta^{13}C$ in the Atlantic in contrast to the more negative $\delta^{13}C$ in the Pacific. Between 4 and 2.5 Ma, however, the gradient between Atlantic and Pacific gradually weakened (Fig 10), indicating penetration of low $\delta^{13}C$ waters from the

Antarctic-controlled deep circulation into the Atlantic. The slow rise of $\delta^{13}C$ in the Pacific could result from a relatively high flux of Northern Component Water into the Antarctic, flowing toward the Southern Ocean shallower than the northerly Antarctic abyssal flow below it. The global SST warm transient between ~5.4 to 3 Ma (Herbert et al, 2016), as well as evidence for elevated atmospheric $CO_2$ suggests an important perturbation of the carbon cycle. The SST perturbation is also reflected in the bulk and planktonic foraminiferal stable isotopes at Sites U1338 and 573 (Reghellin et al, 2015; Drury et al.,

2018). At Site 1241 on the Cocos Ridge northeast of the sites in this paper, records of temperature and biogenic MAR find the interval between 5.4 and 3 Ma to be unusually warm but with relatively low biogenic production when compared to the LMBB (Seki et al., 2012).

The movement of the equatorial Pacific locus of high biogenic deposition from the east central equatorial Pacific to east of

the Galapagos (Fig 1; Farrell et al, 1995; Lawrence et al, 2006) may have triggered PPLC-4, by removing $CaCO_3$ rain from the region west of the Galapagos Islands that had until then supported higher $CaCO_3$ burial at the sea floor. However, low $CaCO_3$ MAR apparently marks the PPLC-4 interval in the easternmost Pacific as well (Sites 846 and 847, Farrell et al, 1995) and gives evidence that moving the locus of production is not the primary cause for PPLC-4.

Bell et al (2015) have given a good summary of stable isotopes in the Atlantic and far South Atlantic, which they compared to Site 849. Sites U1338 and U1337 have virtually the same $\delta^{13}C$ signature as Site 849 (Drury et al, 2016, 2017; Tian et al, 2018) so Site 849 is representative of an east-central Pacific deep water signal. The trend in the Pacific $\delta^{13}C$ time series rises somewhat between 5 and 3.6 Ma, but the trend toward higher $\delta^{13}C$ stops around then (Fig 10). During the period between 4.4 and 4.2 Ma, the southernmost South Atlantic sites (Site 704) became isotopically heavier and more like the North Atlantic,

and were interpreted to indicate deep expansion of the NADW tongue and somewhat stronger NADW flow during the warm Pliocene (Hodell and Venz-Curtis, 2006; Bell et al., 2015). After 4 Ma, expansion of an Antarctic source that better mixed Atlantic and Pacific signals penetrated more effectively into both the Atlantic and Pacific to make the Atlantic isotopically lighter and the Pacific heavier. Similarly, Klevenz et al (2008) interpreted Nd isotopes and Cd/Ca on Walvis Ridge to

indicate that the Antarctic end member signal reached a local minimum around 3.5-4 Ma and concluded that NADW flow was highest then. We suggest that the more thorough exchange of Pacific deep water after 3.5 Ma helped to increase deep dissolution in the Atlantic and enhanced Atlantic 100-kyr cyclicity that was beginning to occur in the Pacific.

In the absence of conclusive evidence, we suggest that the enhanced dissolution in PPLC-4 marks the effects of enhanced Antarctic flow within the deep Pacific and a larger reservoir of low $[CO_3]^=$ water in the deep ocean between 4.1 and 2.9 Ma. Sites U1337, U1338, and Site 849 all are sites within the mixing zone of Pacific deep water outflow with relatively weak deep flow from the Antarctic, at least compared to the western Pacific. The lack of strong change of $\delta^{13}C$ despite apparent increases in Antarctic influence worldwide, suggests continued deep DIC storage in the Pacific. Since the period between 5

and 3.5 Ma was also warm (Herbert et al, 2016), there may also be feedbacks from the carbon cycle as well as enhanced shelf storage of $CaCO_3$ from high sea levels reducing deep $CaCO_3$ burial. However, sea levels were relatively high for millions of years prior to the 4-3 Ma period of PPLC-4 (Rohling et al, 2014; Miller et al, 2005), so the sea level effect should not have been large.

**6 Conclusions**

We have developed site-to-site correlations at the meter scale and a composite age model for 7 ODP and IODP drillsites from the eastern equatorial Pacific in order to study the evolution of eastern equatorial Pacific sedimentation. This fulfils one of the objectives of the Pacific Equatorial Age Transect, and the high-resolution stratigraphy provides a foundation that should benefit future regional studies.

We identified five long-term low $CaCO_3$ intervals within the 7 drill sites we investigated: PPLC-5 (4737-4465 ka), PPLC-4 (3 intervals between 4093 and 2915 ka), PPLC-3 (2 intervals on either side of a $CaCO_3$ high, between 2684 and 2248 ka), PPLC-2 (2135-1685 ka), and PPLC-1 (402-51 ka). With bulk chemical data and the geographic range of the investigated drill sites it is possible to distinguish between dissolution and production as causes of low $CaCO_3$ intervals in the Pliocene-Pleistocene record. We found that PPLC-5 and PPLC-2 result from $CaCO_3$ dilution through diatom production, and the other

3 result from enhanced $CaCO_3$ dissolution

The magnitude of $CaCO_3$ dissolution can be described by changes in the CCD. However, local and regional variability of sedimentation makes the CCD method only useful for intervals of large carbon cycle change and/or low time resolution. The ratio $CaCO_3:BaSO_4$ makes it possible to study dissolution at an individual drill site. We used this ratio to show that 100-kyr

dissolution cycles were regionally strong by 1600 ka, in contrast to oxygen isotopes which didn't develop 100-kyr cyclicity until ~1000 ka. We also used $CaCO_3:BaSO_4$ to confirm that the highest $CaCO_3$ dissolution occurred in PPLC-4, between 4.1 and 2.9 Ma.

It is possible that PPLC-4 resulted from closure of the CAS, but there are complicating factors, like the shift of the locus of $CaCO_3$ burial eastward in the Pacific at the end of PPLC-5, the last high deposition interval of the LMBB. High flushing of the Atlantic basin is indicated by the Atlantic-Pacific $\delta^{13}C$ gradient, but the gradient was actually weakening in the 4-3 Ma interval suggesting more Antarctic DIC storage or more total DIC in the ocean. New work concludes that atmospheric $CO_2$ was declining in this interval as well (Stap et al, 2016) and is also associated with declining but global SST was still high (Herbert et al, 2016). The multiple changes in climate and carbon cycle indicators show that significant reorganizations were occurring in the early Pliocene despite a relatively stable ice volume indicated by benthic oxygen isotopes. The data we have presented here suggests that regional changes are strong and will be critical to understand the transition to the beginning of Northern Hemisphere glaciation.

## Data Availability

All data for this paper are available in the Supplemental Material as well as at PANGAEA (https://Pangaea.de) as the PDI-21251 data set.

## Author Contributions

ML provided XRF data and calculations of $CaCO_3$ from bulk density. AJD and TW provided stable isotope data and tuned age models. JT provided additional stable isotope data to improve the age model. RW and JT provided splices for the IODP Expedition 321 drill sites, while RW checked and revised splices for the ODP Leg 138 drill sites. ML and AJD wrote most of the original text while all authors participated in the revisions.

## Competing Interests

The authors declare that they have no conflicts of interest.

## Acknowledgments

We thank all the members of IODP Expedition 320/321 and those from ODP Leg 138 (Scientists, technicians, drillers, and ships crew) who collected the basic information and sediments needed for this study. Lyle was funded by NSF grant OCE-0962184 and grant OCE-1656960.  Westerhold and Drury were funded by the Deutsche Forschungsgemeinschaft (DFG) grant WE5479-1 and WE5479-3 and Drury was a postdoctoral researcher in H. Pälike's ERC Consolidator Grant EARTHSEQUENCING (grant agreement 617462). Jun Tian was funded by National Science Foundation of China (Grant No. 41525020, 41776051) and Program of Shanghai Subject Chief Scientist (A type) (16XD1403000).

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

**Tables**

Table 1: Drill Sites investigated in this study

| Site | Latitude (°N) | Longitude (°E) | Water depth (mbsl) | 5 Ma Latitude (°N) | 5 Ma longitude (°E) | Data Available |
|---|---|---|---|---|---|---|
| ODP 848 | -2.994 | -110.480 | 3854 | -3.77 | -106.09 | GRA-CaCO$_3$, this paper |
| ODP 849 | 0.183 | -110.520 | 3838 | -0.60 | -106.25 | scanning XRF, GRA-CaCO$_3$, this paper |
| ODP 850 | 1.297 | -110.521 | 3786 | 0.51 | -106.30 | GRA-CaCO$_3$, this paper |
| ODP 851 | 2.770 | -110.572 | 3760 | 1.98 | -106.41 | GRA-CaCO$_3$, this paper |
| IODP U1338 | 2.508 | -117.970 | 4206 | 1.43 | -113.79 | scanning XRF, Lyle et al (2012); Lyle and Backman (2013); Wilson (2014) |
| IODP U1337 | 3.834 | -123.206 | 4468 | 2.55 | -119.07 | scanning XRF, Wilson (2014) |
| IODP U1335 | 5.312 | -126.284 | 4333 | 3.92 | -122.19 | scanning XRF, Wilson (2014) |

Table 2 Plio-Pleistocene Low Carbonate Intervals (PPLCs)

| Low carbonate interval | Age Interval (ka) | Interval length kyr | Marine Isotope Stage | Magnetic Chron | Depth Site U1338 (m CCSF) | Depth Site 849 (m CCSF) | Depth Site 848 (m CCSF) | Depth Site 850 (m CCSF) | Depth Site 851 (m CCSF) | Depth Site U1337 (m CCSF) | Depth Site U1335 (m CCSF) | Notes |
|---|---|---|---|---|---|---|---|---|---|---|---|---|
| PPLC-01 | 51-402 | 351 | 4-11 | C1n | 1.06-5.14 | 1.71-12.71 | 0.81-5.53 | 1.17-8.52 | 0.78-4.07 | 0.79-5.87 | 0.55-2.68 | Most prominent at Site 849; carbonate cycles obscure it to the west |
| PPLC-02 | 1685-2135 | 450 | 59-81 | lower C1r.2r to base C2r.1r | 22.20-28.77 | 48.77-61.38 | 25.02-27.70 | 34.98-42.71 | 31.93-39.64 | 25.21-31.73 | 10.13-12.10 | See also Povea et al, 2016 for interval in Sites 1238 and 1240; matches 849 |
| PPLC-03a | 2248-2383 | 193 | 87-93 | C2r | 29.23-31.56 | 62.61-67.23 | 28.15-29.68 | 43.81-47.91 | 39.92-42.11 | 32.27-34.81 | 12.27-13.32 | two parts with prominent CaCO$_3$ high between. |
| PPLC-03b | 2532-2684 | 152 | 100-G1 | C2r/C2An.1n | 33.59-35.71 | 71.10-74.94 | 30.84-31.61 | 51.13-54.70 | 45.17-47.34 | 36.50-39.30 | 14.12-14.98 | Prominent CaCO$_3$ high in between the two |
| PPLC-04a | 2915-3372 | 457 | G16-MG4 | lower C2An.1n to upper C2An.3n | 39.19-45.84 | 81.14-93.52 | 33.20-35.37 | 60.44-69.81 | 52.91-61.06 | 43.07-51.79 | 16.33-18.39 | The CaCO$_3$ low gets stronger away from equator |
| PPLC-04b | 3558-3731 | 173 | MG12-Gi 11 | lower C2An.3n to upper C2Ar | 48.24-50.85 | 97.68-102.36 | 35.83-37.12 | 73.45-77.31 | 64.46-66.95 | 54.34-57.93 | 19.04-19.68 | The CaCO$_3$ low gets stronger away from equator |
| PPLC-04c | 3834-4093 | 258 | Gi16-Gi 24 | lower C2Ar | 52.42-56.35 | 105.69-112.51 | 37.54-38.68 | 80.02-85.25 | 68.95-72.75 | 60.45-67.45 | 20.06-21.18 | The CaCO$_3$ low gets stronger away from equator;PLC-04b and 04c join in U1337, but are distinct in U1335, U848 |
| PPLC-05 | 4465-4737 | 272 | N2-NS 6 | base C3n.1r to lower C3n.2r | 62.69-69.33 | 123.91-143.00 | 40.35-44.27 | 94.10-107.72 | 80.42-87.55 | missing | missing | Very prominent at equator, missing away from it |

**Figures**

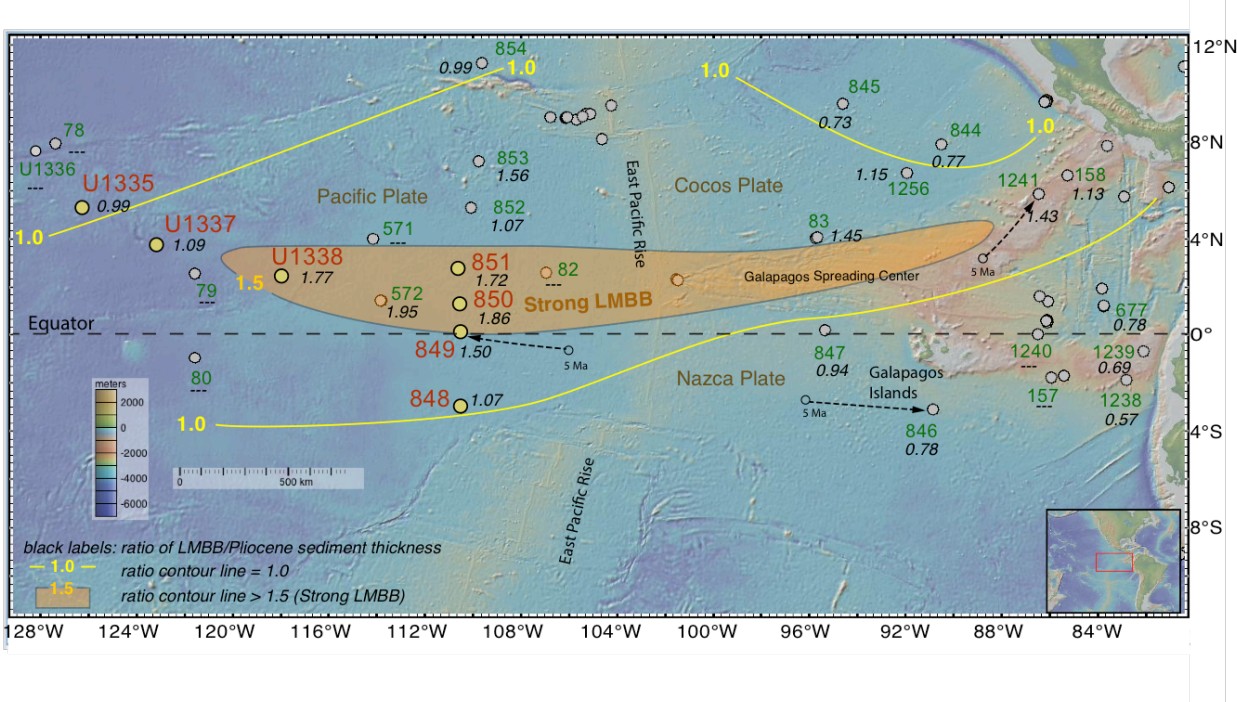

Figure 1: Overview of East Pacific drill sites from GeoMapApp (http://www.geomapapp.org/). Gray dots indicate positions of DSDP/ODP drill sites. Red site labels: ODP and IODP drill sites from the eastern equatorial Pacific used in this paper with other DSDP/ODP/IODP Sites labeled in green. All sites are at their modern locations. Example backtracked positions at 5 Ma are shown for Site 849 on the Pacific Plate (representative for U1335, U1337, U1338, and Sites 848-854), Site 846 on the Nazca Plate (representative for Sites 847, 1238, and 1239), and Site 1241 on the Cocos Plate (representative for Sites 844, 845, and 1256), assuming a hotspot reference frame. Thickness of the 8.5-4.5 Ma sediment divided by the 4.5-0 Ma sediment thickness (LMBB/Pliocene ratio) are indicated by the italic black labels at each drill site. Sites on crust younger than 8 Ma, or where the section was incompletely drilled, or with a hiatus in the interval are marked with "---". Yellow lines mark where the LMBB/Pliocene ratio is 1, while shaded orange region is where the ratio is >1.5, marking the major LMBB locus of deposition. Thickest LMBB sequences are found on sites with crust older than 8 Ma on the Pacific and Cocos Plates just north of the equator. Sites south of the equator in the easternmost Pacific have thicker sediments in the 4.5-0 Ma time range, indicating high young sedimentation rates.

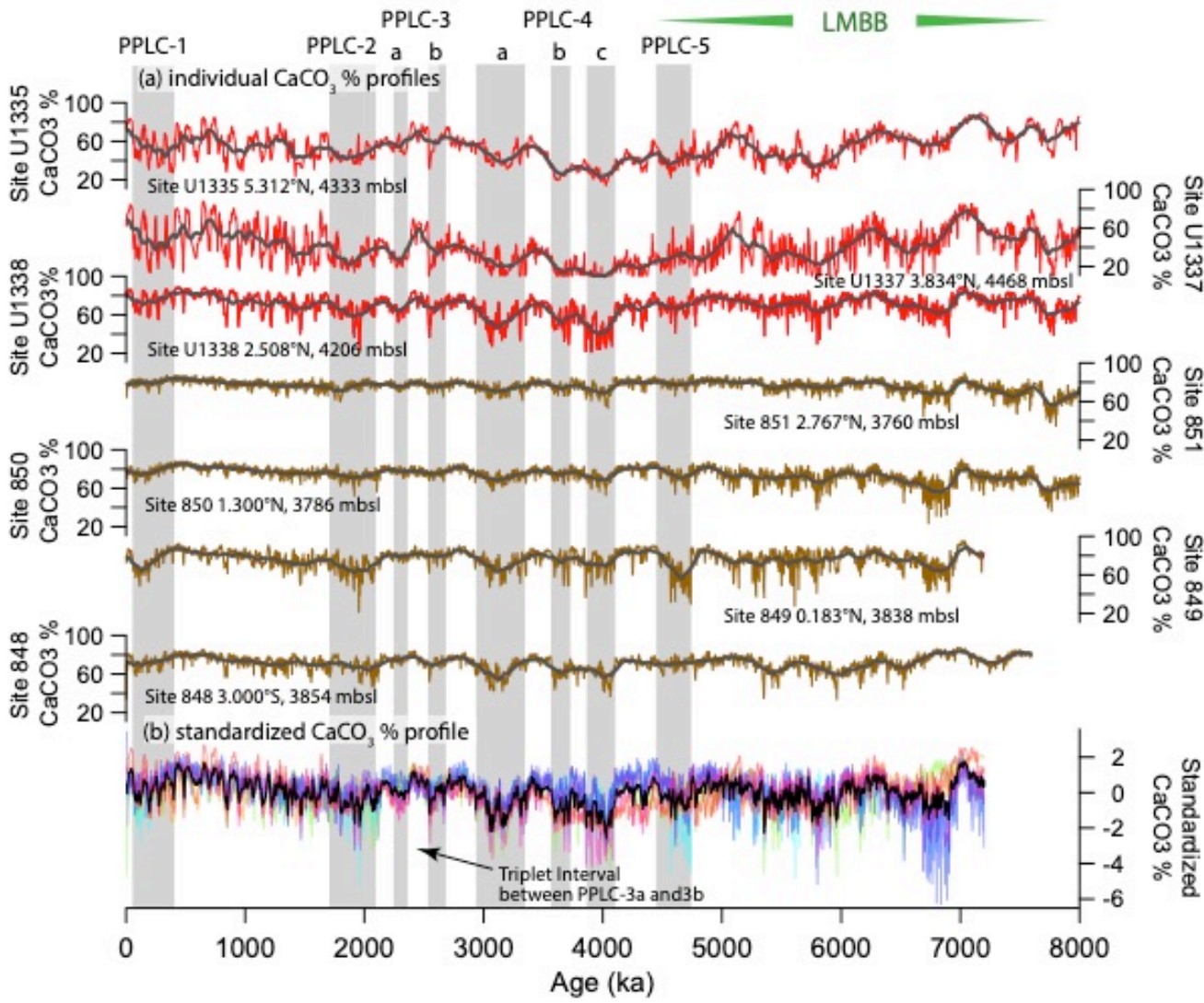

**Figure 2: (a) CaCO₃ wt % profiles from the 7 drill sites in this paper. Red profiles are from sites drilled by IODP Exp 320/321 while the brown profiles are sites drilled on ODP Leg 138. Black lines are the smoothed profiles. The gray shading marks PPLC intervals determined in this paper (See Table 2), while the hatched intervals are productivity controlled. LMBB is the Late Miocene Biogenic Bloom and includes PPLC-5, actually early Pliocene in age. (b) Stacked and standardized CaCO₃ % profiles. Each profile is expressed in standard deviations from the mean and the data from each drill site has a different color. The black profile is the stack of all the CaCO₃ % curves. The PPLCs were determined by common low CaCO₃ % in the profiles.**

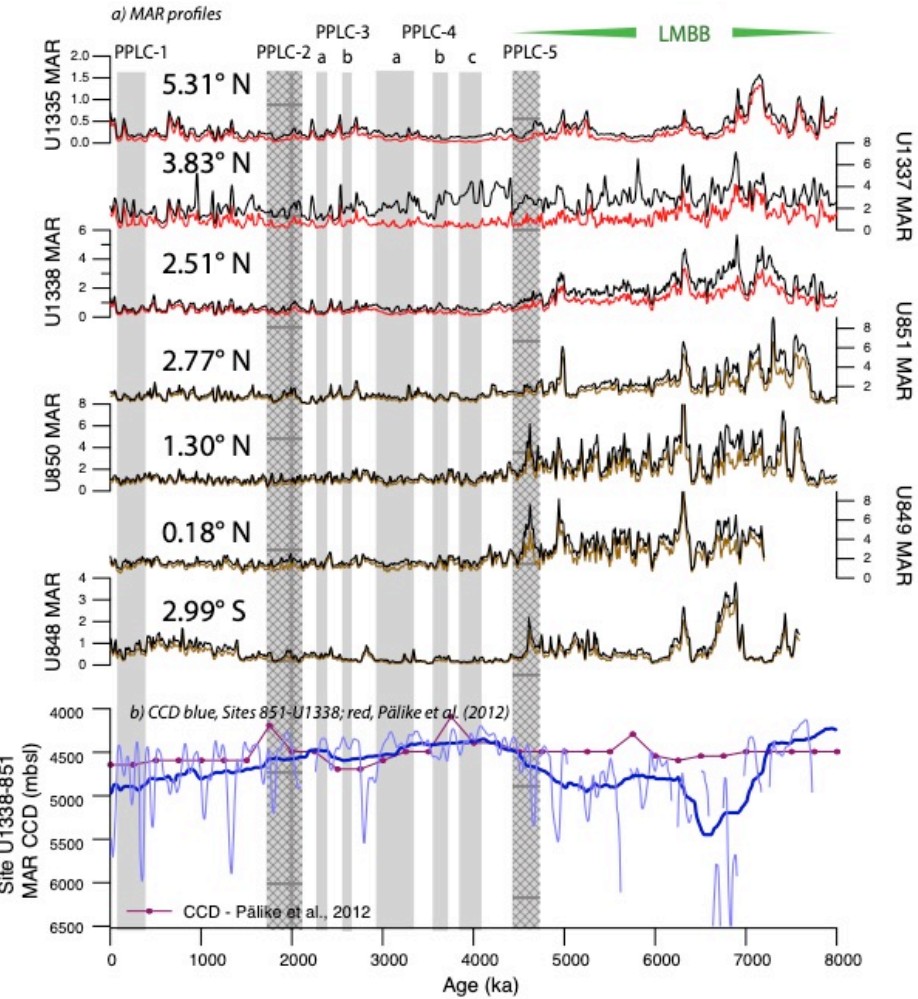

**Figure 3: (a) Mass Accumulation Rates (MARs) for eastern equatorial Pacific drill sites. Bulk MAR (black) and CaCO₃ MAR (brown—Leg 138; red—Exp 320/ I21) profiles from drill sites studied for this paper. Shaded intervals are PPLCs, and shaded hatched intervals are productivity controlled. MAR data are at 10 kyr intervals. Sites are arranged from south to north, at their modern position. Only Site U1337 has bulk MAR time series that diverge significantly from the CaCO₃ MAR profiles, indicative of sediment focusing at Site U1337. (b) Eastern Pacific CCD change from Pälike et al (2012; red line) and using the gradient of deposition from Site 851 (3760 m) to Site U1338 (4206 m) with two levels of smoothing, 50 kyr (light blue line) and 750 kyr (heavy blue line). Breaks in the light blue line profile are intervals where the CaCO3 MAR at the deeper Site U1338 exceeded that at Site 851. The 50-kyr smooth finds transient CCD changes up to 1.5 km, but probably contains significant noise from local sediment variability. The 750-kyr smooth displays how the CCD was shallowest between 4200 and 3100 ka, during the time of PPLC-4.**

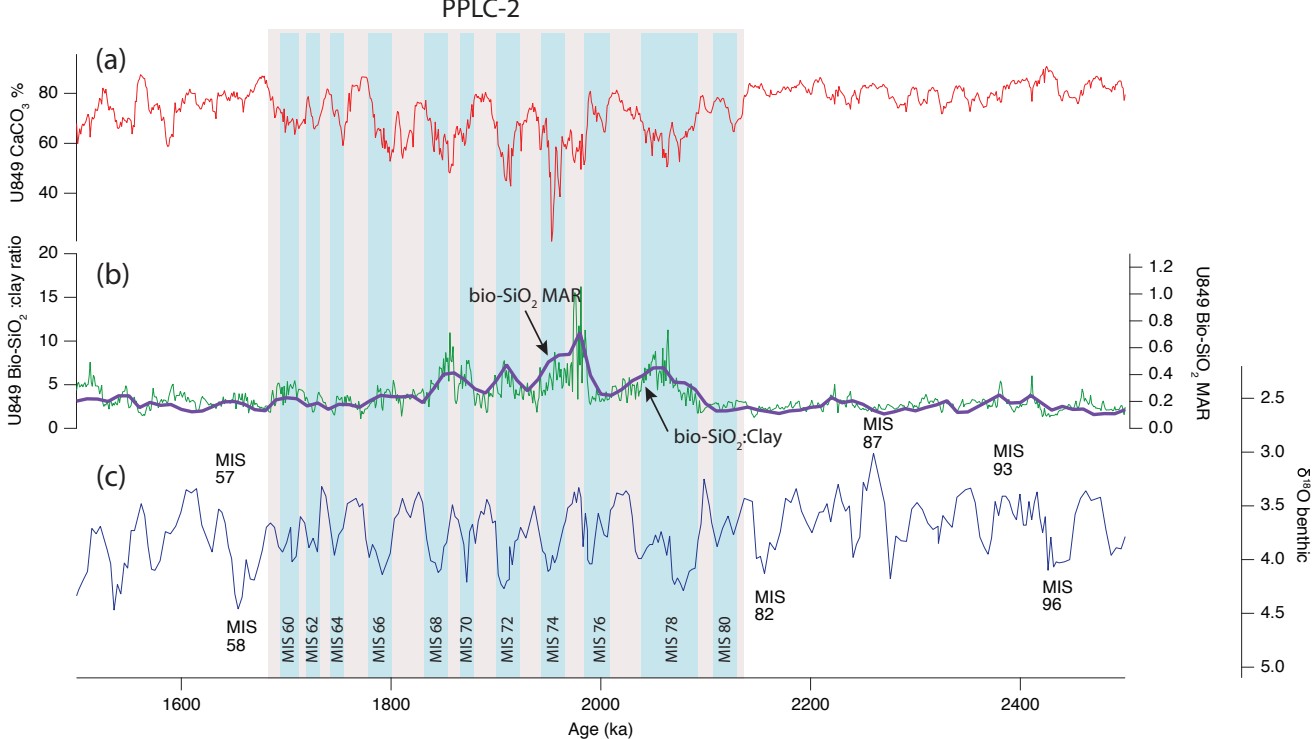

**Figure 4: Site 849 profiles over low carbonate interval PPLC-2 (2135-1685 ka) showing bio-SiO$_2$ dilution as a primary cause of the low CaCO$_3$ %. High bio-SiO$_2$ MARs are in glacial intervals. (a) Scanning XRF CaCO$_3$%; (b) Thick line is bio-SiO$_2$ MAR, and thin line is bio-SiO$_2$:clay ratio, showing the high bio-SiO$_2$ MAR associated with each sub-interval in PPLC-2. (c) Site 849 benthic oxygen isotope stratigraphy constructed with data from Mix et al. (1995) using the updated splice from this paper. Stages are numbered following Lisiecki and Raymo (2005). Glacial intervals are labelled.**

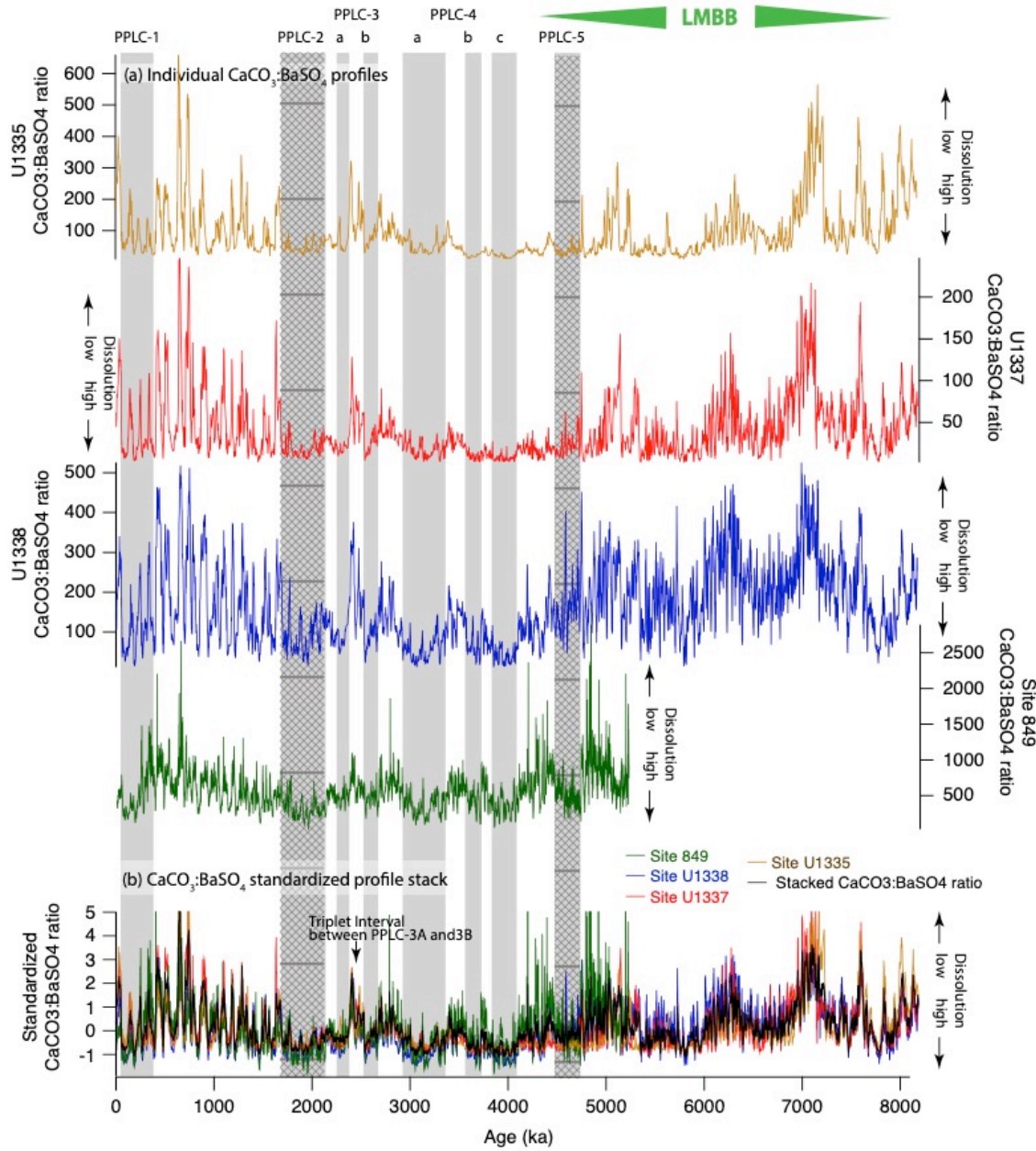

**Figure 5: XRF CaCO$_3$:BaSO$_4$ profiles to investigate CaCO$_3$ dissolution at drill sites with scanning XRF profiles. Shaded intervals are PPLCs and shaded hatched intervals are productivity driven. Low CaCO$_3$:BaSO$_4$ indicates high dissolution of CaCO$_3$. (a) Individual profiles of CaCO$_3$:BaSO$_4$ at Sites U1335, U1337, U1338 and Site 849. Note how ratios are higher at Site 849, the shallowest site nearest to the equator. (b) Changes in dissolution in the eastern Pacific shown by stacked CaCO$_3$:BaSO$_4$ records expressed as standard deviations from the average. Low CaCO$_3$:BaSO$_4$ coincides with PPLC's, indicating that the dissolution signal has an important impact on the record, even within high productivity intervals.**

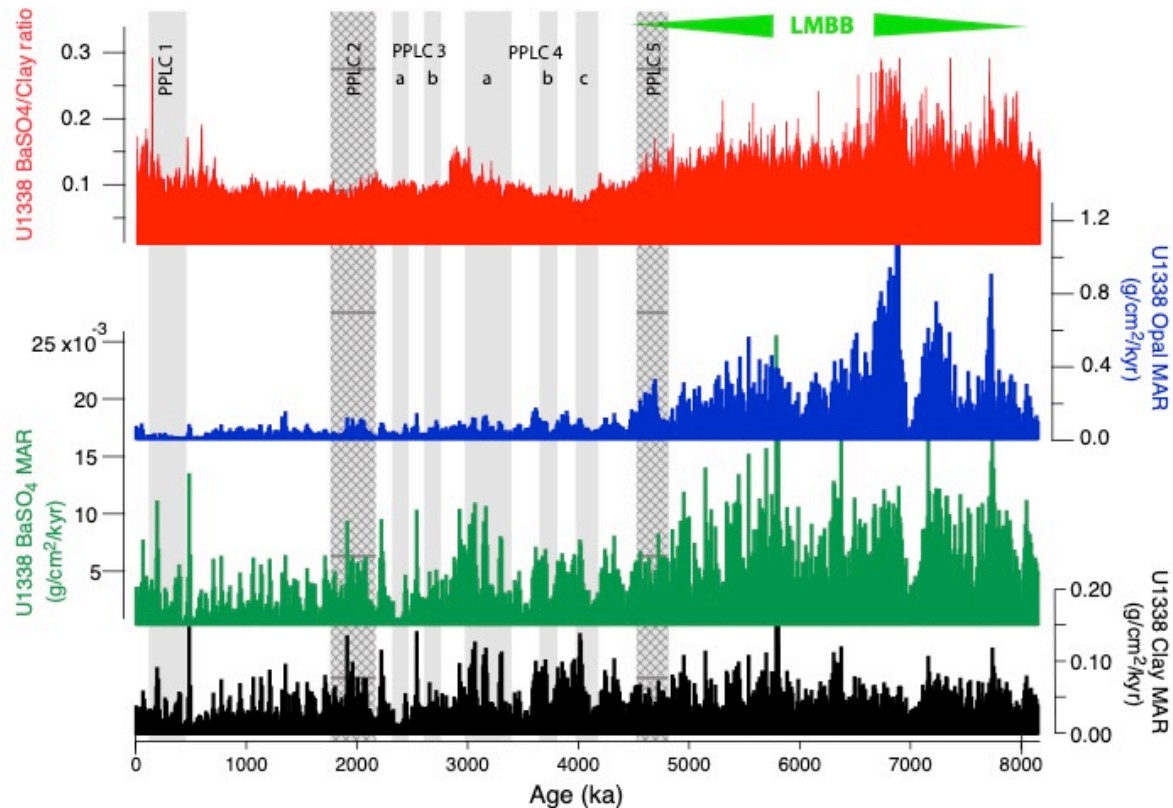

**Figure 6: Site U1338 BaSO₄/clay ratio and Mass Accumulation Rates (MAR) of biogenic sediment components and clay. Gray bands mark PPLCs (Pliocene-Pleistocene Low CaCO₃ intervals) and hatched gray bands are productivity controlled. High bio-SiO₂ and BaSO₄ MAR prior to 4465 ka represents the LMBB interval at Site U1338. Clay MAR is relatively constant over the 8 million years and is shown by the drop in BaSO₄/clay ratio at that time.**

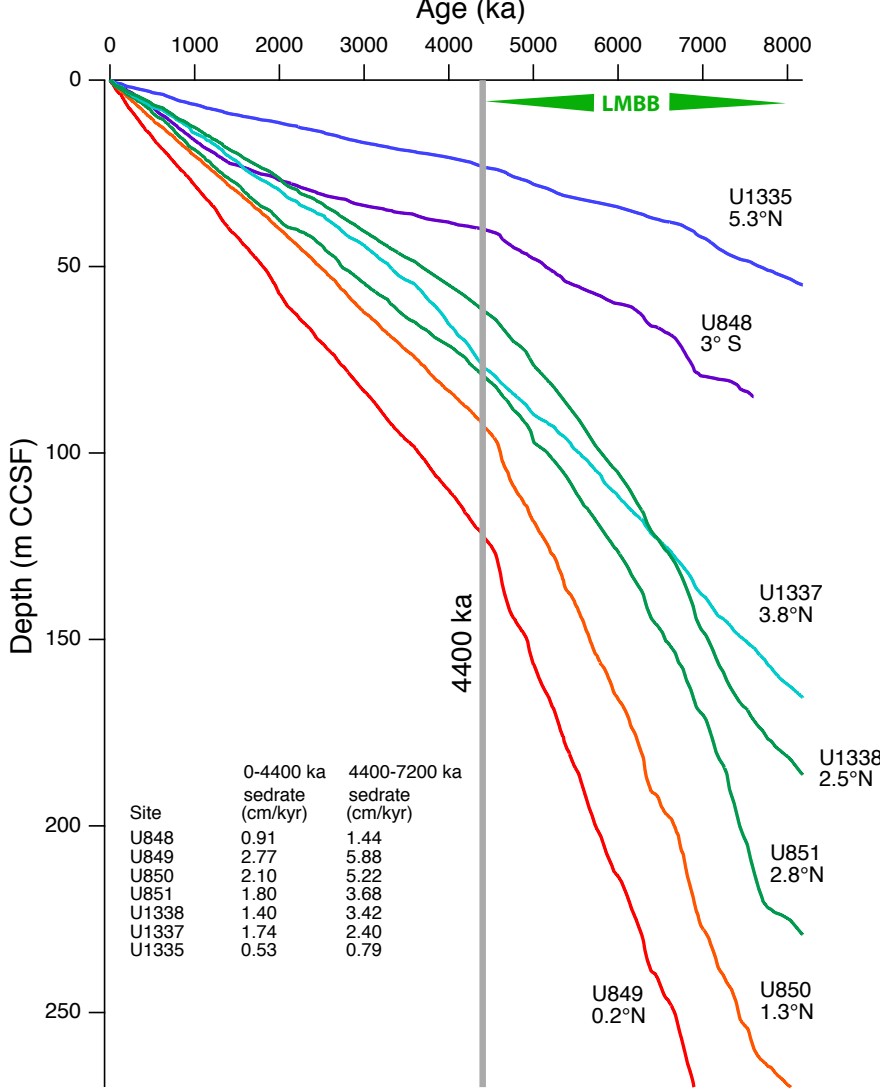

**Figure 7: Age-depth curves for ODP Leg 138 Sites 848, 849, 850, and 851 with IODP Expedition 320/321 Sites U1335, U1337, and U1338, based on age model in this paper. Steeper slopes represent faster sedimentation rates. Steepest slopes are found at sites that are nearest to the high productivity equatorial upwelling zone. Average sedimentation rates for the section younger than 4400 ka and for rates between 7200 ka (base of XRF section scanned in Site 849) and 4400 ka are tabulated. The break in sedimentation at the gray line at 4400 ka marks the end of the LMBB. CCSF (Composite Coring depth below Sea Floor) is the depth within the continuous spliced sediment section**

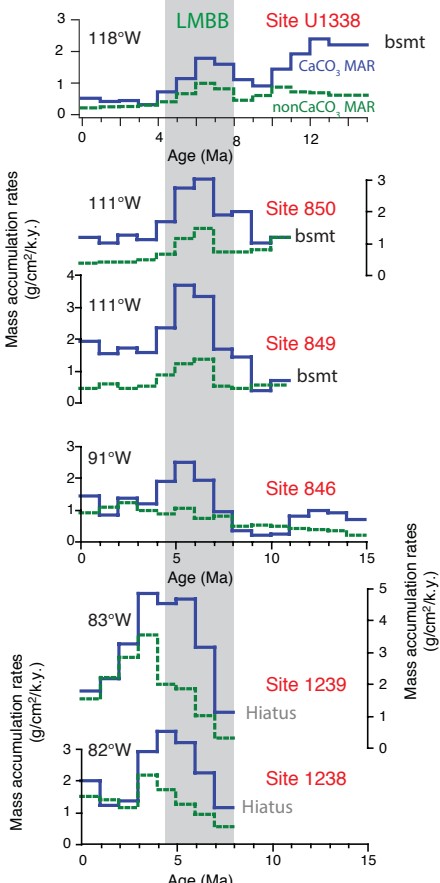

**Figure 8: 1 Myr increments of CaCO₃ MAR (blue lines) and non-CaCO₃ MAR (Green dashed lines) on a longitudinal equatorial transect from 118°W to 82°W, near Ecuador. Non-CaCO₃ deposition is mostly biogenic silica at these drillsites. Basement is marked by "bsmt". Data for U1338 is from this paper, while the other data are from ODP Leg 202 Shipboard Scientific Party (2003). The 8-4.5 Ma LMBB window is shaded. The end of peak CaCO₃ deposition is roughly 2 million years younger near South America than in the west. Eastern Pacific drill sites (Sites 846, 1238, and 1239) have increased biogenic deposition post-LMBB, presumably because of nutrient concentration after the Central American Seaway closed.**

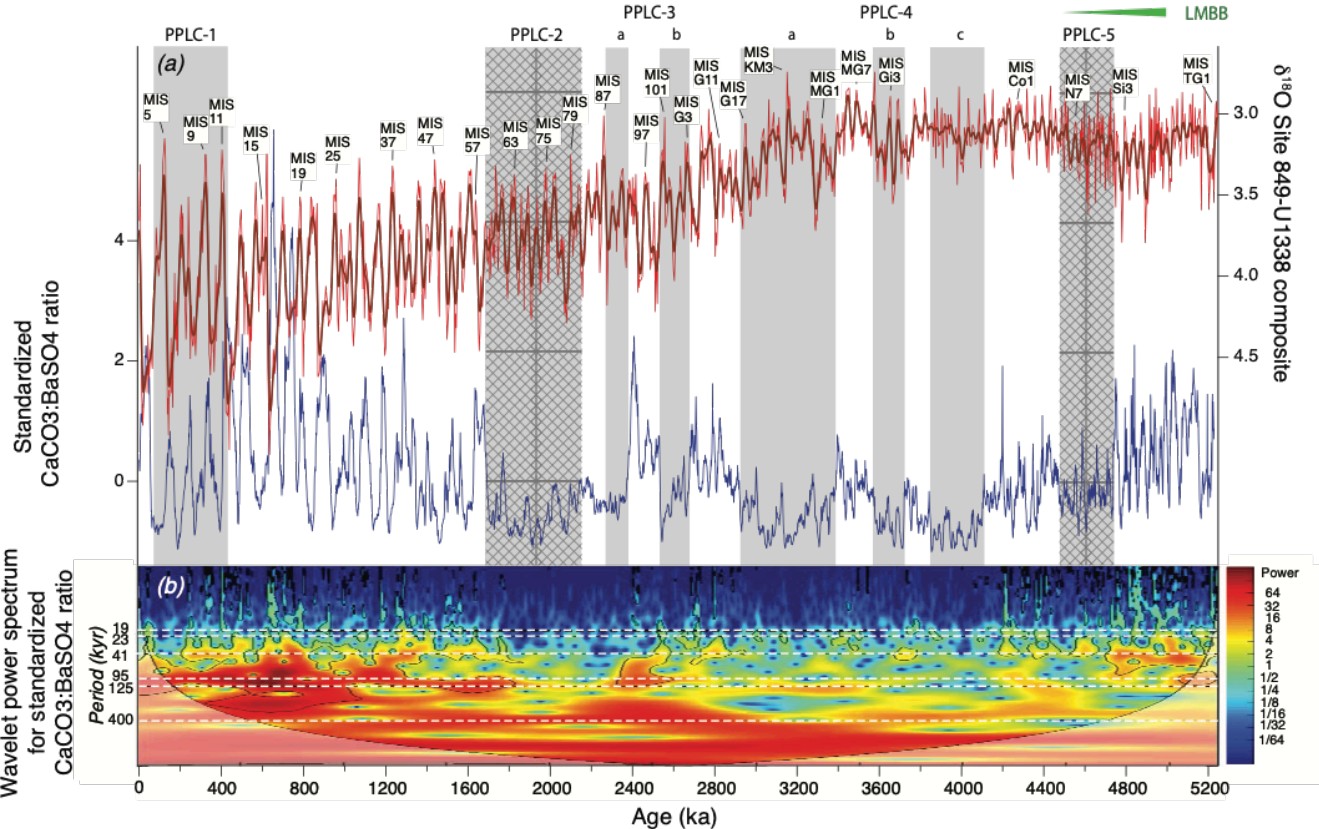

**Figure 9: (a) A comparison of the stacked CaCO₃:BaSO₄ record of dissolution cycles (blue line) to the combined U1338-849 benthic oxygen isotope record (red and brown lines). The thick brown line on the oxygen isotope record is a 7 point binomial smooth of the data. A strong 100-kyr periodicity to the dissolution cycle begins roughly at MIS 58, with low dissolution associated with periods of high benthic δ¹⁸O (high ice volume). Before MIS 58, the association is less clear. (b) Wavelet analysis of the CaCO₃:BaSO₄ stack, showing 400 kyr power throughout the CaCO₃:BaSO₄ record and development of 100 kyr power in the interval beginning about 1800 ka and continues to the Holocene.**

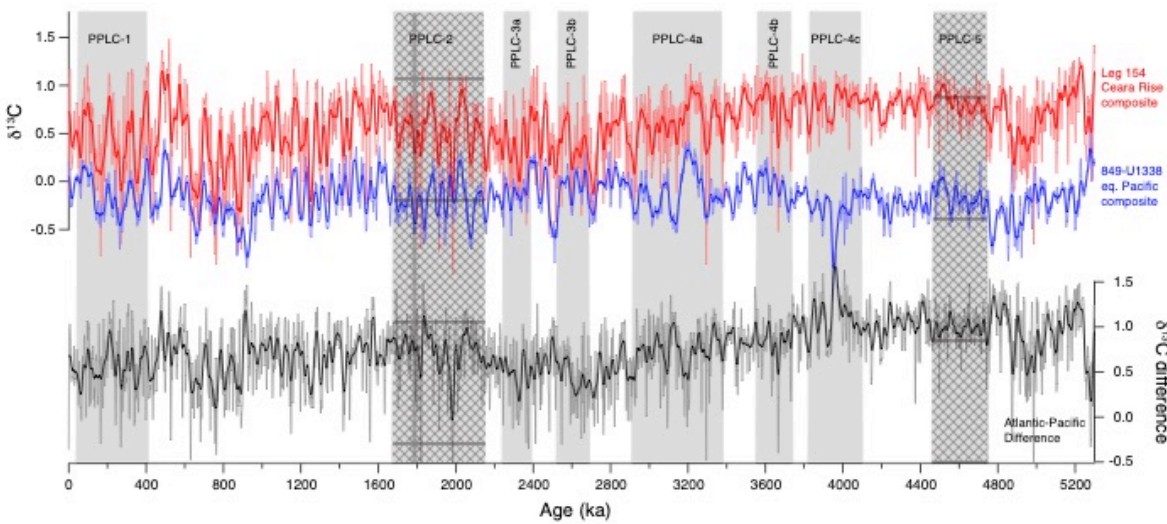

**Figure 10: δ¹³C records from Ceara Rise (Atlantic, red) and the Site 849-U1338 composite (Pacific, blue), and the Atlantic-Pacific δ¹³C difference (black). The PPLC-4 dissolution interval falls on the period of decline of the Atlantic Pacific difference caused by a decrease in Atlantic δ¹³C of about 0.7‰ between 4000 ka and 2500 ka, and an increase in Pacific δ¹³C of about 0.4‰ over the same interval.**