# Peer review of "Late Miocene to Recent High Resolution Eastern Equatorial Pacific Carbonate Records: Stratigraphy linked by dissolution and paleoproductivity"

_Climate of the Past, 2018_

## Referee Comment (RC1) · Anonymous Referee #1 · 3 Jan 2019

General comments

The manuscript of Mitchell Lyle et al. "Late Miocene to Recent High Resolution Eastern Equatorial Pacific Carbonate Records: Stratigraphy linked by dissolution and paleoproductivity" is supposed to be published in "CPD". In their study, the authors critically discuss causes for the observed $CaCO_3$ deposition in the Eastern Equatorial Pacific over the past 8 Myr, i.e., production vs. dissolution. The study is based on XRF-derived bulk sediment composition data and mass accumulation rates from sites of IODP Expedition 320/321 and ODP Leg 138. The major outcome of the study is apparently the

identification of five long-term low CaCO3 intervals within the past 8 Myr – two of them as a result of CaCO3 dilution through diatom production, and the other 3 as a result of enhanced CaCO3 dissolution.

The overall story presented is based on an innovative approach and has the potential to be published in CPD. However, I recommend this paper for publication only with the revisions described below. Most importantly, the abstract and introduction lack a working hypothesis and a few sentences on the overall aim of the study. The discussion is thus not easy to follow in various parts and needs some re-structuring.

Specific comments

Abstract: I won't start the abstract with "We report. . .". To me, there are 1-2 sentences missing at the beginning of the abstract summarizing the aim of your study (i.e., what are the scientific questions you want to solve).

Page 1 Lines 12-14: Include information on locality of study sites, i.e., EEP.

Introduction (p. 1, l. 27 to p. 3, l. 7): In this part of the introduction you should make clear what's the aim of your study, i.e., what are yet unresolved scientific questions that you want to answer or what is your working hypothesis. From the introduction as it is now, the aim of your study is not clear to me.

Lines 5-7 and 26-29: These are results and therefore should not be part of the introduction.

Lines 9-12: Add the location where your records are from. You want to update the stratigraphy from 0-5.3 Ma, but work on the 0-8 Myr period. What about the 5.3-8 Myr stratigraphy? This should be mentioned here. "We choose 5.3-0 Ma because it has good age control": To me that's not an argument. You can mention that as an additional point but not as a reason of why this interval has been selected.

Lines 20-23: This is no information relevant for the introduction. Please move this section together with l. 18-20, p. 4 into section 3.2. This will also avoid repetitions. To me it is also not clear why to generate a combined age model for all sites instead of using the original age models? Could you briefly explain that in the "age model section"?

Line 1-3: Include in parenthesis the ODP and IODP sites, respectively.

Lines 3-7: I think this information is not relevant to your story, please delete.

Line 9: Change "All 7 sites have continuous orbital-resolving records of estimated CaCO3..." to "We generated (correct?) continuous orbitally-resolved records of CaCO3 for all 7 sites..."

Lines 9-16: From this section it is not clear to me which records are new, and which records are already published. Please rephrase. Also, why are you mentioning records that go back to 24 Ma while you are only studying the 8-0 Ma interval?

Lines 21-33: I suggest to move these sections to line 8.

Lines 12-13: Sites 848 and 850 where tied to the U1338-tied Site 849? Is that correct? Why were Sites 848 and 850 not also tied to U1338? And why have both Sites U1338 and 849 been selected as alignment sites?

Lines 2-3: I don't understand: Are these already published data (because of the references) or are these new data?

Line 7: "Unpublished opal analyses": If you use these data to calibrate something, you

have to publish them in this paper. Provide them at least in the supporting information.

Section 4: After the methods I expect the results, in which you just describe the data you have generated (i.e. sedimentation rates range from x-y, CaCO3 fluctuates between x and y, MARs vary between x and y, etc.). However, Section 4 is more a discussion. Restructure sections 4-6 accordingly, i.e., either provide a section for the results and another section for the discussion, or provide a combined "results and discussion" section with several sub-sections. I would prefer the latter in order to avoid repetitions.

Line 19-21: I don't understand.

Lines 31: "(1) the percentage profiles can be directly measured as the sediment sections are processed": This is not an argument.

Line 1: "(3) the results are easily compared to earlier observations at other cores and drill sites": I don't get that point.

Lines 3-5: Delete, because the same information is also given in the next sentence.

Lines 7-8: I think there is a third important factor you should also consider: What about a simple decrease in CaCO3 production due to, e.g. Fe limitation, and thus reduced phytoplankton productivity?

Line 14: Is there a better word for "defects"? What are the "defects"?

Lines 17-23: To me this paragraph does not belong to this section. Move to Section 6.2. Is the cyclicity of CaCO3% described in this paragraph based on a visual evaluation or have you done some phase analysis (I guess you only show a wavelet analysis of the CaCO3:BaSO4 ratio)? "CaCO3 % is high in Pleistocene glacial climate intervals": Is the temporal resolution of your age model sufficient to determine whether CaCO3%

is highest during peak glacial conditions (as written now) rather than during glacial terminations as observed in previous studies?

Section 5.2: Why does such an anomaly not also occur at nearby Sites U1335 and U1338? Are they too far away from Site U1337?

Lines 9-10: Any explanation for that observation?

Line 1: This means that opal MAR is unrelated to glacial-interglacial change? It would be nice to highlight glacial or interglacial intervals in the figure.

Line 21: I guess you mean three peaks by "CaCO3 triplet", correct? However, I cannot see them in the figure. Make sure that your figures are large enough in size and/or highlight special features such as the CaCO3 triplet.

Lines 10-15: This paragraph should go into the introduction section where you talk about the CCD.

Lines 3-9: I can't follow. Please rephrase.

Line 11: You estimate the depth of the CCD based on CaCO3 MAR. The latter you get from equation (1). But how do you transfer CaCO3 MAR into the depth of the CCD? Also, can you provide a data table where you show all the values inserted into equation (1) to get CaCO3 MAR? Provide link to figure.

Section 6.1: How do your CCD estimates compare to that of Pälike et al. (2012)? Could you plot them together? Their CCD is somewhere between 4.2 and 4.8 km for your study interval. How can you explain this difference? Also you get a depth of the

CCD of >6 km, how realistic is that?

Lines 15-16: "In other words, attributing PPLC-4 to one cause is overly simplistic." I don't understand, please rephrase.

Section 6.4/Lines 25-26: I would not finish a paper like this. Since the paper is very long, I would delete the entire section.

Table 1: Change "Drill Sites in this study" to "Drill sites investigated in this study". Add ODP and IODP to Sites. "Length of dated record": Is that information relevant for your study? If not, please delete. "Data available": Are these data sets that are already available and were worked on in this study (then references are missing), or are these data sets that were generated in this study? Please rephrase respectively.

Table 2: "MIS at Site 849": Delete "at Site 849". Magnetic Chron: Not mentioned in the text. I would delete that column.

Figure 1: Start caption with something like "Overview of East Pacific drill sites". "Sites U1335, U1337, U1338, and 849 have XRF scanning chemical data, while Sites 848, 850, and 851 have CaCO3% estimated from bulk density": This information is not relevant for the figure. Delete. LMBB/Pliocene ratio: Not mentioned in the text. So if relevant, provide a discussion on this ratio in the main text, if not, delete.

Figure 3: This figure has not been mentioned in the main text; this doesn't work. As I understand, the only aim of this figure is to show the higher temporal resolution of your new records compared to previously published data of Lyle and Baldauf (2015). So this figure does not contribute to the story presented in your paper. I suggest either to delete this figure, or to move it into the supporting information, but including the data from Lyle and Baldauf as a comparison.

[Figure]

Figure 4: a) I guess the black lines are a smooth, and the red and brown lines are raw data. Please add this information to the figure caption. b) Different colors refer to different sites? Please explain.

Figure 5: "MAR data are at 10 kyr intervals": Delete, but make sure that this information is given in the methods. "Sites are arranged from south to north, at their modern position": Delete. b) Maybe I missed it in the text: Why are these two sites used for CCD estimations? Please add information to the main text. "with two levels of smoothing": Can you please also show the unsmoothed record? Delete "The 50 kyr smooth...time of PPLC 4". I suggest to remove the CCD record from that figure and to show it in a separate figure together with the CCD record from Pälike et al. (2012).

Figure 6: b) Use different colors for opal MAR and opal:clay. c) Are these only the Mix et al. (1995) data? I don't get that from the figure caption. Please rephrase accordingly.

Figure 7: To make it easier for the reader, I suggest to add arrows, indicating that low (high) CaCO3:BaSO4 represents high (low) CaCO3 dissolution. Is it possible to fill the 5-8 Ma gap in the CaCO3:BaSO4 record? If no data for Site 849 available, then only based on the records from the remaining sites? Then you can also extend the records shown in Fig. 8 back to 8 Ma.

Figure 8: See my comment to Fig. 7. a) Explain solid and thick lines. b) Enlarge labeling of "power" and "period". Make dashed "period-lines" more prominent.

Figure 9: Explain solid and thick lines.

Technical corrections:

Several abbreviations are not introduced. Make sure to introduce all of them.

Versus vs vs. Remain consistent.

Bio-SiO2: Do you mean biogenic SiO2? Please rephrase.

Labeling of sites: ODP Sites without "U", IODP Sites with "U". Please correct accordingly.

Line 12: change "late Miocene-Recent" to "late Miocene to recent"

Line 14: Add ODP

Line 15: Delete "The period between" and move "4.5 and 8 Ma" to "LMBB" in the sentence before.

Line 20: Change "We use a combined age model..." to "We use a combined age model for all sites investigated by joining...". Delete "data from".

Line 22: Add reference to "At 5 Ma, the sites span from ∼4°S to ∼4°N".

Line 2: Delete "these".

Line 1: I guess Site 849 is missing in the heading.

Lines 11 and 17-18: Remove, because this information is already given on p. 5, l. 2-3.

Lines 15-17: Change "Unfortunately, Hagelberg et al (1995) did not publish their CaCO3 estimates for the Leg 138 Sites, so we redid the estimate for Sites 848, 850, and 851 along the revised splices presented here" to "Here we provide CaCO3 estimates for Sites 848, 850, and 851 along the revised splices".

Line 2: Remove "unfortunately".

Line 12-13: Repetition. Delete.

Line 24: Change "CaCO3 MAR records..." to "Except for Site U1337, CaCO3 MAR records..." and remove "however" in the following sentence.

Line 26: Change "...that is evidence..." to "... that is evident..."

Lines 27-28: Change "Most of the drill sites in this paper have variability in the bulk sediment MAR, but most of the bulk MAR variation is typically derived from changes CaCO3 MAR. Site U1337 is unique..." to "Variability in bulk sediment MARs of the sites investigated in this study is typically derived from changes in CaCO3 MAR. However, Site U1337 is unique..."

Line 2: Change "in addition to" to "our interpretation becomes supported by".

Line 12: Add "that we observe" to "intervals".

Line 23: Change "farther" to "further".

Lines 25-27: Provide link to figure.

Lines 29-30: Delete "We infer...CaCO3 content".

Line 20: Delete ":"

Line 22: What do you mean by "they"?

Line 9: Change "glacial carbonate cycles" to "glacial-interglacial carbonate cycles".

Line 31: Delete "for".

Lines 19-20: This information should be part of the figure caption and not of the main text.

Lines 31-34: Repetition of p. 13, l. 4-8. Please restructure.

Line 2: Change "...XRF scanning data are available, there..." to "...XRF scanning data are available (i.e., Sites XXX, YYY; ZZZ), there..." and remove "at the four drill sites with XRF data" (l. 5-6).

Lines 30-34: This doesn't work. You already provide a link to Fig 8 earlier. Delete this sentence, but make sure that the information about the smoothed d18O record is given in the figure caption.

Lines 2-3: Change "the correlation is still strong when CaCO3:BaSO4 is compared to the smoothed oxygen isotope record" to "that correlation is still strong".

Line 12: Add "Site" to 607.

Lines 6-9: Delete these sentences.

Lines 1-2: Delete "In this paper...work regionally".

Lines 4-9: Restructure as follows: "We identified five long- term low CaCO3 intervals within the 7 drill sites we investigated: PPLC-5 (4737-4465 ka), PPLC-4 (3 intervals between 4093 and 2915 ka), PPLC-3 (2 intervals on either side of a CaCO3 high, between 2684 and 2248 ka), PPLC-2 (2135-1685 ka), and PPLC-1 (402-51 ka). With bulk chemical data and the geographic range of the investigated drill sites it is possible to distinguish between dissolution and production as causes of low CaCO3 intervals in the Pliocene-Pleistocene record. We found that PPLC-5 and PPLC-2 result from CaCO3 dilution through diatom production, and the other 3 result from enhanced CaCO3 dissolution."

Line 16: Delete "whose 3. . .and 3 Ma".

Line 21: Studies from 2016 are not really "new". Replace "new" by "previous".

Line 7: Remove "Late Miocene Biogenic Bloom".

Line 7: Change "atandardized" to "standardized".

―――――――――――――――――――

---

## Referee Comment (RC2) · Reghellin (Referee) · 10 Jan 2019

General Comments

I enjoyed reading and reviewing the Lyle et al. manuscript (MS No.: cp-2018-157). For most of the part, it is a well written and structured manuscript, which presents high resolution sediment properties and compositional records from multiple locations of the eastern equatorial Pacific Ocean. The Results and their interpretation represent a fundamental contribution to the effort of understanding how this region and, more in

general low latitude ocean areas, works as the ocean/climate system evolved to modern conditions. The Discussion mainly focuses on five Plio-Pleistocene intervals of low carbonate content. The causes of low carbonate content in the EEP, dilution by biosilica particles versus dissolution of carbonate particles, are here examined using carbonate content, MAR of different sediment components, sedimentation rates, CaCO3:BaSO4 ratios and benthic foraminifera stable isotopes records from multiple locations of the EEP.

The main issues I found in the manuscript mostly concern parts of the Methods and Results and are listed in "Specific comments". The inaccuracies typical of carbonate content estimations from GRA density have not been discussed. It is not clear how CCD depth was calculated from CaCO3 MAR and data are missing. In the Results/Discussion it is not always clear how intervals of low carbonate content were interpreted as reflecting dilution by biosilica rather than greater carbonate dissolution or vice versa. The Biogenic bloom interval is improperly defined and its proposed end is imprecise. Given this, my overall evaluation is that this manuscript has the potential to be published in Climate of the Past, but I recommend to revise the manuscript according to all the comments presented below before publication.

Specific comments

- Abstract. At the beginning of the Abstract, add a couple of sentences introducing the problem(s) at the base of this study and the reasons for this study. Also, add a reference to the study area. There is no reference of the EEP in the whole Abstract.

- Introduction. As for the Abstract, in the Introduction I couldn't find the unresolved questions that stay at the base of this study. Please add them together with the aims of this study.

- Biogenic bloom. I think it is incorrect to refer to the Biogenic Bloom as "Late Miocene Biogenic Bloom (LMBB)" because this event did not occur only during the Miocene as it ended in the early Pliocene. This is stated in the manuscript several times and is

reported in the literature. For example, the studies cited on page 7 line 13 (Farrell et al., 1995, Lyle and Baldauf, 2015) refer to the biogenic bloom as "Biogenic Bloom" or as "late Miocene early Pliocene biogenic bloom". In the paper it is also awkward to read that an early Pliocene interval (e.g. PPLC-5) is part of the LMBB. For example, on page 35 line 4 what is defined as a late Miocene event (LMBB) includes another event from the early Pliocene. It makes little sense to me. I strongly recommend to substitute "late Miocene Biogenic Bloom (LMBB)" with "Biogenic Bloom" or with "late Miocene early Pliocene biogenic bloom" throughout the text, figures, figure captions and tables. Also change all the acronym "LMBB" accordingly, throughout the text, figures, figure captions and tables.

- Data presented. In the text it is not so easy to keep track where the presented data come from (a previous study, this study, etc.) because of the important amount of data. It would make things easier for the reader to add some columns to Table 1 or even adding a new table in which is clearly listed type of data, drill site, data origin and time span of all the data presented in the study.

- CaCO3 % estimates from GRA density (sections 3.4 and SM 5). There is no mention in the manuscript that carbonate content estimates calculated from GRA measured in EEP sediment have been demonstrated to lead to imprecise carbonate estimations. Reghellin et al. (2013) [Reghellin, D., G. R. Dickens, and J. Backman (2013), The relationship between wet bulk density and carbonate content in sediments from the Eastern Equatorial Pacific, Marine Geology, 344, 41-52] have shown that it is not possible to accurately describe the relationship between carbonate content using a single equation, like it was done in this study. The estimation errors are particularly evident at high CaCO3 content (>60 %; the case of most of the records presented here) because of the wide range of WDB. Fig. SM-4 show that at Site 849 CaCO3 estimation from GRA, XRF and discrete measurements give similar values but still differences are apparent between GRA and discrete measurements in the figure. On the basis of the results presented by Reghellin et al. (2013), new carbonate content records estimated from

GRA density cannot be published without fully considering the uncertainties and inaccuracies of the method. In addition, the relationship varies significantly in the EEP from a site to another because of sediment composition differences (carbonate vs biosilica content and, within carbonate components, the type of them) at different locations. In the SM only one power law equation (5) is presented so I have the doubt that it was computed using all discrete CaCO3 and density data from Sites 848, 849, 850 and 851 together. This can potentially bring to even greater errors in the CaCO3 estimates than using different power law equations at different sites. If this is the case, I recommend to recalculate CaCO3 estimates using different power law equations at different sites. Each power law equation should be computed using the data from that particular site only.

- MAR data (section 3.5). I tried to calculate some MAR for a few sediment components using the equation (6), with dry bulk density, sedimentation rates and CaCO3 % data in the SM tables and the results were different from corresponding MAR data listed in your tables. Please verify if data presented in the tables are correct.

- Biogenic bloom end. On page 7 line 16 is stated "is easily observed by the change in slope of the age-depth curves at about 4400 ka (Fig. 2)". To me is evident that the most significant and clear change in the age-depth curves of Fig. 2 is a bit to the right of where the grey line is placed, which is at about 4600 ka. This is particularly evident in curves of Site U848, U1338, U851, U850 and U849. I would then strongly recommend to: i. move the end of the biogenic bloom line at 4600 ka in Fig. 2; ii. move green arrow tip to 4600 ka; iii. recalculate average sedimentation rates for the periods 0-4600 ka and 4600-7200 ka in Fig. 2; iv. modify all figures, figure captions and manuscript text according to the above reasoning.

- Sediment focusing (page 10 line 6). "Surprisingly, the sediment. . . . . .strongly affect the CaCO3 % profile" isn't this because of moderate currents strength preferentially removes light-fine sediments as you say on lines 1 and 2? If this is the case it makes sense to me that sedimentation rates increase, CaCO3 % decreases but leaving the

CaCO3 % profiles unaltered. And CaCO3 MAR would increase because of higher sedimentation rates, even if dry density decreases because of the input of fine sediments. This is not anomalous to me.

- Dilution vs dissolution. In the Results (sections 5.3-5.5) it is often missing a clear explanation of how it was distinguished between carbonate dilution by biosilica and carbonate dissolution. Section 5.3. The first paragraph (page 10 lines 12-19) is difficult to follow and needs rewriting because it is not clear how you interpreted some intervals (i.e. PPLC-5 and PPLC-2) as high production rather than high carbonate dissolution. Is it because carbonate% is low at on equator sites and not as much at off equator sites? or is it because of low carbonate% and relatively high carbonate MAR at on equator sites? The CaCO3:BaSO4 used to estimate carbonate dissolution is also very low at PPLC-2 at all sites (Fig. 7). Couldn't it be that high production also generated greater carbonate dissolution? How is it possible to completely exclude carbonate dissolution in this interval? I recommend to clearly state in which way it is possible to distinguish between dilution and dissolution. Section 5.4. Lines 14-15: it is not clear how data at PPLC-4 indicate dissolution rather dilution by biosilica. Provide clear explanation. Section 5.5. Here it is easier to follow your rationale because you introduce the CaCO3:BaSO4 proxy. I strongly recommend to add references to this proxy in the interpretation of the other PPLC intervals (so in sections 5.3 and 5.4). It would make your interpretation of low carbonate intervals much stronger and easier to follow by the reader.

- CCD depth estimate from CaCO3 MAR (section 6.1). It is not clear how MAR CCD were calculated. From reasoning in 6.1 it seems that you have calculated MAR CCD from CaCO3 MAR using equation (1) in page 13, but in that equation there is no MAR CCD. Clarify this. I could not find the data used to calculate MAR CCD anywhere in the manuscript nor in SM. I strongly suggest to add a table listing all data used to make curves in Fig. 5 panel b. While reading section 6.1 I got the feeling that the estimation of CCD depth from CaCO3 MAR is a weak approach. This is because you

state that i) "local sediment anomalies.........cause significant noise to this approach", ii) "weaker signal in the Pliocene and Miocene is harder to distinguish from noise", iii) the CCD estimate "suffers from the noise resulting from building the trend with records from only 2 sites" and iv) "minor errors in correlation.........are magnified in the CCD estimate". Plus, it is not clear how you calculated CCD depth from CaCO3 MAR. Also, I see there are significant differences between the CCD depth record (Fig. 5 panel b) and the equatorial Pacific CCD of Pälike et al. (2012). The latter ranges between about 4200 and 4700 mbsl whereas yours mostly between 4300 and 5500 mbsl. How can you explain this difference? It seems to me that all the issues of the method are strongly affecting the CCD estimates. Isn't it enough to speculate on carbonate preservation (dissolution) over time by using the CaCO3:BaSO4, CaCO3 MAR, CaCO3 % and sedimentation rates? In my opinion the message emerging from this part of the Discussion would be much stronger without the CCD depth estimate.

- Page 17 lines 22-24. Add that evidence of early Pliocene higher SST and lower biogenic production compared to during the biogenic bloom comes also from bulk sediment stable isotopes and sedimentation rates records from Leg 138 on-equator sites and from Site 573 and U1338 (see Shackleton and Hall., 1995 and Reghellin et al., 2015) [Reghellin, D., H. K. Coxall, G. R. Dickens, and J. Backman (2015), Carbon and oxygen isotopes of bulk carbonate in sediment deposited beneath the eastern equatorial Pacific over the last 8 million years, Paleoceanography, 30, doi:10.1002/2015PA002825].

- Page 18 lines 5-6. Add explanation of how the "expansion of an Antarctic source.........and Pacific" can cause opposite changes in the d13C records from the Atlantic and the Pacific

Technical corrections

Page 1 Line 12: define "XRF" Line 13: define "IODP" and "ODP" Line 23: define "DIC"

Page 2 Line 2: add "Reghellin et al., 2013" to references. Line 25: define "DIC" Line

27: add references Line 28: define "NADW" Line 34: substitute square brackets with round brackets.

Page 3 Lines 4-7: move this sentence to Results or Discussion Lines 13-15: add references Line 26: define "XRF" Line 31: define "IODP"

Line 32: define "ODP" and "GRA"

Page 4 Line 2: change "3°S" to "3.0°S" Line 11: define "ccsf". Also, "ccsf" is in the text sometimes written as lowercase and sometimes as uppercase. Modify to be consistent

Page 5 Line 11: add explanation of the criteria used to choose the base or master drill site

Page 6 Line 5: define "ICP-MS" Line 14: add "Reghellin et al., 2013" to references

Page 7 Lines 16-17: substitute "4400 ka" with "4600 ka" Line 20: define "DSDP"

Page 8 Line 3: late Miocene carbonate crash here defined from 11 to 8 Ma and on page 7 line 11 from 10 to 8 Ma. Change time periods to be consistent Line 5: in References list there are two Pälike et al., 2010b. To which one do you refer? References list

Page 9 Lines 3-8: the same concept is repeated in these two sentences. Keep one and delete the other. I would also add low carbonate production as a cause for low CaCO3 intervals.

Page 10 Line 26: add "(Figure 3)" after "at Site U1338"

Page 11 Line 1: define "MIS" Line 14: substitute "lowest record" with "panel b" Lines 15-19: these are speculations. Move to Discussion Line 22: substitute "bottom" with "panel b"

Page 12 Lines 27-30: add references

Page 13 Line 25: is (1) a novel equation? If so state it otherwise provide reference

Page 15 Line 5: how about PPLC-2? I see that also during this time period the

CaCO3:BaSO4 is very low at XRF sites Line 32: specify the type of smoothing

Page 16 Line 10: define "AABW"

Page 17 Line 5: substitute "implies" with "would imply" Line 11: define "NCW" Line 15: define "SST"

Page 18 Line 31: add "early Pliocene" after "late Miocene" Line 32: add "Reghellin et al., 2015"

Page 19 Line 34: define "CAS"

Page 27 There are two Pälike et al., 2010b. Fix

Page 31 Table 1: change "length (ka) of dated record" to "length of dated record (ka)". Can you specify what is meant with "Data available"? See comment in "Specific comments"

Page 32 Specify origin of background map. In the map are present many light grey dots which (drill sites locations?) that are not labelled. Add a label or remove the dots

Page 33 See comment "Biogenic bloom end" in Specific comments section Figure 3: subscript 3 in "CaCO3" in panel a scales Substitute "4400 ka" with "4600 ka" in figure and figure caption In figure caption, define what is the grey vertical line

Page 35 What are darker curves on panel a? is it a kind of smoothing of actual data record? Clarify in figure caption Line 4: see "biogenic bloom" comment in "Specific comments"

Page 36 Can you revers y axis scale on panel b? With shallower depths at the top it is difficult to follow lines 5-6: see "CCD depth estimate from CaCO3 MAR" comment in "Specific comments" line 6: specify which curve is the 50 kyr smoothing (blue line?) and which one is the 750 kyr smoothing (bold blue line?)

Page 37 Please add a sentence in figure caption about alignment of d18O glacial

intervals and high opal MAR

Page 39 Specify in figure caption what the dark red line on the stack isotope curve is

Page 40 Specify in figure caption what are dark color lines

SM Page 1 Lines 26-27: given what is written here I do not understand why in the captions of Tables SM-1 to SM-4 (ODP sites) you refer to CSF and CCSF depths and in captions of Tables SM-5 to SM-7 (IODP sites) you refer to mbsf and rmcd depths. Please correct depth scales in all supplemental tables or clarify

SM Page 2 Line 16: define "IMPH"

SM page 5 Line 24: substitute "since they have partially..." with "because they have partially..." Line 25: define "APC" Line 26: define "XCB"

SM page 6 Lines 30-31: describe method used to obtain GRA estimates

SM page 7 Lines 8-9: can you state where the "discrete CaCO3 data" come from? and can you clarify with which data the power law estimate (5) was computed? Line 23: specify the unit of "Xe" Line 24: change "grams" to "g/cm3"

SM page 8 Lines 19-20: Can you provide a possible cause of this sediment accumulation/erosion feature at Site U1337? Lines 22-23: "Site U1337......significantly elevated relative to the CaCO3 MAR" add reference to Fig. 5

SM page 12 On panel d there are two marks for Site 572 location. Fix

SM page 14 Change "Figure SM-1" to "Figure SM-3"

SM page 15 Change "Figure SM-2" to "Figure SM-4"

Daniele Reghellin

---

## Author Comment (AC1) · 8 Mar 2019

**Response to reviewers of "Late Miocene to Recent High Resolution Eastern Equatorial Pacific Carbonate Records: Stratigraphy linked by dissolution and paleoproductivity**

5    Mitchell Lyle, Anna Joy Drury, Jun Tian, Roy Wilkens, and Thomas Westerhold

*In this response the original critique is in black and responses by the authors to the reviews are written in blue. We thank the reviewers for their thorough critiques of our paper "Late Miocene to Recent High Resolution Eastern Equatorial Pacific Carbonate Records:  Stratigraphy linked by dissolution and paleoproductivity".  We have revised the paper substantially in response to the comments. We have added information to the abstract to*
10  *better explain the objective of the paper. We have rearranged and rewritten the introduction with this goal in mind as well. We have  recombined the original section 4 with the discussion so it is now below results.  And we have followed advice to clarify the writing and add material to the supplemental material.  One major addition was a better explanation of how the CaCO3 MAR gradient method worked to determine the CCD, and tables of the CCD determination by this method, as well as estimates of paleodepth to the sea floor through time. Because*
15  *of the reorganization, figure numbers in the revision have changed order, as listed below:*

| *Figure # in Submission* | *Figure # in Revision* |
|---|---|
| *Fig 1* | *Fig 1* |
| *Fig 4* | *Fig 2* |
| *Fig 5* | *Fig 3* |
| *Fig 6* | *Fig 4* |
| *Fig 7* | *Fig 5* |
| *Fig 2* | *Fig 6* |
| *Fig 3* | *Fig 7 (figure revised and repurposed)* |
| *Fig 10* | *Fig 8* |
| *Fig 8* | *Fig 9* |
| *Fig 9* | *Fig 10* |

30

*RESPONSE TO ANONYMOUS REVIEWER 1*

*Reference R1 requested extensive reorganization, and we felt that the R1 comments had merit. We added sentences to the abstract to define the objectives better, re-organized the introduction to give a broader outlook, and moved section 4 "Eastern Tropical Pacific Sedimentation since 10 Ma" back into the Discussion section. We*

5 *reorganized the figures (see table) to match text changes. We added more discussion and tables to clarify how the CaCO3 MAR gradient was used to establish the CCD. We have also addressed the specific comments in the R1 review, as shown below.*

Anonymous Referee #1

General comments

The manuscript of Mitchell Lyle et al. "Late Miocene to Recent High Resolution Eastern Equatorial Pacific Carbonate Records: Stratigraphy linked by dissolution and paleoproductivity" is supposed to be published in "CPD". In their study, the authors critically discuss causes for the observed CaCO3 deposition in the Eastern

15 Equatorial Pacific over the past 8 Myr, i.e., production vs. dissolution. The study is based on XRF-derived bulk sediment composition data and mass accumulation rates from sites of IODP Expedition 320/321 and ODP Leg 138. The major outcome of the study is apparently the identification of five long-term low CaCO3 intervals within the past 8 Myr – two of them as a result of CaCO3 dilution through diatom production, and the other 3 as a result of enhanced CaCO3 dissolution.

The overall story presented is based on an innovative approach and has the potential to be published in CPD. However, I recommend this paper for publication only with the revisions described below. Most importantly, the abstract and introduction lack a working hypothesis and a few sentences on the overall aim of the study. The discussion is thus not easy to follow in various parts and needs some re-structuring.

25

**Specific comments**

Abstract: I won't start the abstract with "We report: : :". To me, there are 1-2 sentences missing at the beginning of the abstract summarizing the aim of your study (i.e., what are the scientific questions you want to solve). Page 1 Lines 12-14: Include information on locality of study sites, i.e., EEP.

30 *We added 3 sentences to the beginning of the abstract to explain the primary objective of the paper and to highlight the stratigraphic importance as well. We identified the sites as being in the eastern equatorial Pacific.*

Introduction (p. 1, l. 27 to p. 3, l. 7): In this part of the introduction you should make clear what's the aim of your study, i.e., what are yet unresolved scientific questions that you want to answer or what is your working hypothesis. From the introduction as it is now, the aim of your study is not clear to me.

*We revised the introduction to focus the objectives, to understand how CaCO₃ dissolution and production has shaped the regional stratigraphic record. In addition we have added context from the objectives of the Pacific Equatorial Age Transect (PEAT, IODP Expeditions 320 and 321).*

Lines 5-7 and 26-29: These are results and therefore should not be part of the introduction.

*These have been removed from the introduction.*

Lines 9-12: Add the location where your records are from. You want to update the stratigraphy from 0-5.3 Ma, but work on the 0-8 Myr period. What about the 5.3-8 Myr stratigraphy? This should be mentioned here. "We choose 5.3-0 Ma because it has good age control": To me that's not an argument. You can mention that as an additional point but not as a reason of why this interval has been selected.

*The paragraph has been revised to indicate that we have limited the discussion in this paper to the Pliocene and Pleistocene the LMBB has significant regional variation that needs further work to explore properly. A proper investigation of the LMBB must take into account the regional variability, both along the equator and across it.*

Lines 20-23: This is no information relevant for the introduction. Please move this section together with l. 18-20, p. 4 into section 3.2. This will also avoid repetitions.To me it is also not clear why to generate a combined age model for all sites instead of using the original age models? Could you briefly explain that in the "age model section"?

*The sentences were moved. The brief reason that we needed to make a combined age model is that we found missing intervals in the Mix et al (1995) isotope record below where we joined the Site 849 and Site U1338 records. We added sentences to explain this in the text in section 3.2*

Page 4 (section 2)

Line 1-3: Include in parenthesis the ODP and IODP sites, respectively.

*ODP and IODP were added.*

Lines 3-7: I think this information is not relevant to your story, please delete.

*We disagree—for Cenozoic studies it is critical to understand the time-geographic constraints on information from each drill site. We have revised the sentences to reflect the changes in the introduction.*

Line 9: Change "All 7 sites have continuous orbital-resolving records of estimated CaCO3: : :" to "We generated (correct?) continuous orbitally-resolved records of CaCO3 for all 7 sites: : :"

*These paragraphs have been moved to section 3.3 and have been revised to indicate who generated the records.*

Lines 9-16: From this section it is not clear to me which records are new, and which records are already published. Please rephrase. Also, why are you mentioning records that go back to 24 Ma while you are only studying the 8-0 Ma interval?

*These paragraphs have been moved to section 3.3 and are intended to show that additional data is available.*

Lines 21-33: I suggest to move these sections to line 8.

*We have moved the paragraph as suggested.*

Page 5 (section 3)

20  Lines 12-13: Sites 848 and 850 where tied to the U1338-tied Site 849? Is that correct? Why were Sites 848 and 850 not also tied to U1338? And why have both Sites U1338 and 849 been selected as alignment sites?

*We explain this better in the new text in Section 3.1. We found that correlations were less ambiguous by comparing Sites 848 and 850 first to Site 849 and then using the site to site correlation between Site U1338 and Site 849 to interpolate equivalent U1338 equivalent depth than to directly try to correlate those 2 sites to Site*

25  *U1338. We have added text to this effect in Section 3.1.*

Page 6 (section 3)

Lines 2-3: I don't understand: Are these already published data (because of the references) or are these new data?

*Section 3.3 has been revised to better indicate the sources of the data. The intent of this paper is to synthesize the*

30  *data.*

Page 7 (section 3)

Line 7: "Unpublished opal analyses": If you use these data to calibrate something, you have to publish them in this paper. Provide them at least in the supporting information.

*The Site 849 bio-SiO$_2$ analyses were included in the paper as Table SM-21 in the original draft. We have revised the text to make it clear that the data is in the Supplemental Material .*

Section 4: After the methods I expect the results, in which you just describe the data you have generated (i.e. sedimentation rates range from x-y, CaCO3 fluctuates between x and y, MARs vary between x and y, etc.). However, Section 4 is more a discussion. Restructure sections 4-6 accordingly, i.e., either provide a section for the results and another section for the discussion, or provide a combined "results and discussion" section with several sub-sections. I would prefer the latter in order to avoid repetitions.

*Section 4 was moved to be after the results where it fits better. It has been incorporated into the revised paper as Section 5.1*

Line 19-21: I don't understand.

*The language of the section has been revised to make it clearer.*

Page 8 ( old section 5; new section 4)

Lines 31: "(1) the percentage profiles can be directly measured as the sediment sections are processed": This is not an argument.

*This is an important argument.  Much of the science for a drilling expedition happens on board the ship. Most of the low resolution profiles as well as previous CCD reconstructions are based on these shipboard measurements. These shipboard measurements are the basis for Pälike et al (2012) as well. These points were added to the text.*

Page 9 (section 5.1/5.2)

Line 1: "(3) the results are easily compared to earlier observations at other cores and drill sites": I don't get that point.

*Most CCD reconstructions are based upon changes in the CaCO3 % with depth to the sea floor, so of necessity are comparing CaCO3 %.*

Lines 3-5: Delete, because the same information is also given in the next sentence.

*Sentence was removed*

Lines 7-8: I think there is a third important factor you should also consider: What about a simple decrease in CaCO3 production due to, e.g. Fe limitation, and thus reduced phytoplankton productivity?

*You are correct, which is why we worded this as "relative $CaCO_3$ dissolution". We add a sentence to explain*
5   *better.*

Line 14: Is there a better word for "defects"? What are the "defects"?

*This was replaced with "…because of age errors for the timing of bio-events and magnetochrons used for the chronostratigraphy of the earlier age models…"*

Lines 17-23: To me this paragraph does not belong to this section. Move to Section 6.2. Is the cyclicity of CaCO3% described in this paragraph based on a visual evaluation or have you done some phase analysis (I guess you only show a wavelet analysis of the CaCO3:BaSO4 ratio)? "CaCO3 % is high in Pleistocene glacial climate intervals": Is the temporal resolution of your age model sufficient to determine whether CaCO3% is highest
15   during peak glacial conditions (as written now) rather than during glacial terminations as observed in previous studies?

*The 100-kyr carbonate cycles are a well-known feature of the Pleistocene equatorial Pacific, so it is appropriate to discuss this here. This variability is an observation from our records. We discuss timing later in Section 6.2. The statement has been revised somewhat to remove concerns about how we know the timing, but it is based on*
20   *studies ranging back to the Swedish Deep Sea expedition.*

Section 5.2: Why does such an anomaly not also occur at nearby Sites U1335 and U1338? Are they too far away from Site U1337?

*Sediment focusing that we can document tends to be a local phenomenon determined by local topography. We*
25   *document additional evidence for focusing in Supplemental Material Section 7 and Figure SM-5. Since focusing is local, focusing at one site does not determine that there will be focusing in nearby basins. Furthermore, Site U1335 is rougly 400 km to the NW of Site U1337, and Site U1338 is about 600 km away to the SE. They could only be linked by very large-scale regional deposition.*

30   Page 10 (section 5.2/5.3)
Lines 9-10: Any explanation for that observation?

*I added a reference to Lyle et al (2014) that found a similar result that CaCO3 MAR was least affected by sediment focusing in the JGOFS region. Presumably this is because the CaCO3 resides in a larger less transportable fraction.*

5    Page 11 (section 5.3)

Line 1: This means that opal MAR is unrelated to glacial-interglacial change? It would be nice to highlight glacial or interglacial intervals in the figure.

*Glacial intervals over PPLC-2 have been marked in the figure so that it is easier to identify the glacials in different time series. As described in the text, there is an association of opal MAR with glacials but the*
10   *association is noisy.*

Line 21: I guess you mean three peaks by "CaCO3 triplet", correct? However, I cannot see them in the figure. Make sure that your figures are large enough in size and/or highlight special features such as the CaCO3 triplet.

*The triplet has been marked in both figures.*

Page 13 (section 6)

Lines 10-15: This paragraph should go into the introduction section where you talk about the CCD.

*The paragraph was moved to the introduction*

20   Page 14 (section 6)

Lines 3-9: I can't follow. Please rephrase.

*A paragraph was added to the text and the text was revised.*

Line 11: You estimate the depth of the CCD based on CaCO3 MAR. The latter you get from equation (1). But how do you transfer CaCO3 MAR into the depth of the CCD? Also, can you provide a data table where you show

25   all the values inserted into equation (1) to get CaCO3 MAR? Provide link to figure.

*We have added a paragraph to describe the CCD estimate better, and we added a more detailed description and tables to the supplemental material, along with tables of our CCD estimate and paleodepths for Sites U1338 and 851.*

30   Section 6.1: How do your CCD estimates compare to that of Pälike et al. (2012)? Could you plot them together? Their CCD is somewhere between 4.2 and 4.8 km for your study interval. How can you explain this difference?

Also you get a depth of the CCD of >6 km, how realistic is that?

*We added the Pälike et al (2012) estimate to Figure 3 (MAR figure), and explained how the depth to the CCD for Pälike et al (2012) was controlled by a single site (U1334) during the Pliocene and Pliocene that was off axis and not really in the equatorial production regime.  We agree that 6 km is deep, and discussed in the paper how*

5  *local sedimentation vagaries leads to noise in the MAR/depth extrapolation.*

Page 17 (section 6.3)

Lines 15-16: "In other words, attributing PPLC-4 to one cause is overly simplistic." I don't understand, please rephrase.

10  *This has been reworded.  At the present time there is no way to determine the direct link between higher dissolution in PPLC-4 and Central American Seaway closure. Other important changes were occurring in the global carbon system at that time.*

Page 18 (section 6.4)

15  Section 6.4/Lines 25-26: I would not finish a paper like this. Since the paper is very long, I would delete the entire section.

*The section has been reorganized and is now section 5.2. It has been better tied into the objective to understand changes in primary productivity.*

20  Table 1: Change "Drill Sites in this study" to "Drill sites investigated in this study". Add ODP and IODP to Sites. "Length of dated record": Is that information relevant for your study? If not, please delete. "Data available": Are these data sets that are already available and were worked on in this study (then references are missing), or are these data sets that were generated in this study? Please rephrase respectively.

*Changed .*

25

Table 2: "MIS at Site 849": Delete "at Site 849". Magnetic Chron: Not mentioned in the text. I would delete that column.

*We changed "MIS at Site 849" to "Marine Isotope Stage". We retained the Magnetic Chron column because this readily allows correlation to distant drill sites and is standard for Neogene work.*

30

Figure 1: Start caption with something like "Overview of East Pacific drill sites". "Sites U1335, U1337, U1338,

and 849 have XRF scanning chemical data, while Sites 848, 850, and 851 have CaCO3% estimated from bulk density": This information is not relevant for the figure. Delete. LMBB/Pliocene ratio: Not mentioned in the text. So if relevant, provide a discussion on this ratio in the main text, if not, delete.

*Changes to caption have been done. A discussion of this ratio was already in the text; we have now made explicit*
5  *mention of the LMBB/Pliocene ratio.*

Figure 3: This figure has not been mentioned in the main text; this doesn't work. As I understand, the only aim of this figure is to show the higher temporal resolution of your new records compared to previously published data of Lyle and Baldauf (2015). So this figure does not contribute to the story presented in your paper. I suggest
10  either to delete this figure, or to move it into the supporting information, but including the data from Lyle and Baldauf as a comparison.

*This figure has been eliminated, but the data have been repurposed as Fig 7 to show the relative invariance of clay MARs over the last 8 Myr.*

15  Figure 4: a) I guess the black lines are a smooth, and the red and brown lines are raw data. Please add this information to the figure caption. b) Different colors refer to different sites? Please explain.

*This is now Fig 2. In (a) the black lines are now explained in the caption. In (b) we have expanded the description to explain that each drill site is represented by a different color.*

20  Figure 5: "MAR data are at 10 kyr intervals": Delete, but make sure that this information is given in the methods. "Sites are arranged from south to north, at their modern position": Delete. b) Maybe I missed it in the text: Why are these two sites used for CCD estimations? Please add information to the main text. "with two levels of smoothing": Can you please also show the unsmoothed record? Delete "The 50 kyr smooth: : :time of PPLC 4". I suggest to remove the CCD record from that figure and to show it in a separate figure together with the CCD
25  record from Pälike et al. (2012).

*Now figure 3. We think it is important, given that many people do not read the text thoroughly, to reiterate why the MAR data are smoother than the CaCO3 % data.*

Figure 6: b) Use different colors for opal MAR and opal:clay. c) Are these only the Mix et al. (1995) data? I don't
30  get that from the figure caption. Please rephrase accordingly.

*Now figure 4. The figure was changed to have different colors for bio-SiO2 MAR and bio-SiO2:clay. We have*

*also now consistently used bio-SiO2 for biogenic silica. The caption was rephrased about the stable oxygen isotope data.*

Figure 7: To make it easier for the reader, I suggest to add arrows, indicating that low (high) CaCO3:BaSO4 represents high (low) CaCO3 dissolution. Is it possible to fill the 5-8 Ma gap in the CaCO3:BaSO4 record? If no data for Site 849 available, then only based on the records from the remaining sites? Then you can also extend the records shown in Fig. 8 back to 8 Ma.

*(now Fig 5) We added arrows to show direction of low/high dissolution, added the 3-site stack to the 4-site stack record that ended at 5.2 Ma, so that a stacked record now goes back to 8.1 Ma.*

Figure 8: See my comment to Fig. 7. a) Explain solid and thick lines. b) Enlarge labeling of "power" and "period". Make dashed "period-lines" more prominent.

*Labeling was enlarged, and the period lines are more prominent. Thick lines are explained in caption. We disagree with expanding the figure back to 8 Ma because it makes it more difficult to see the changes in the 100-kyr periods in the Pleistocene.*

Figure 9: Explain solid and thick lines.

*The explanation has been added to the figure*

Technical corrections:

Several abbreviations are not introduced. Make sure to introduce all of them. *We believe we have now introduced all abbreviations.*

Versus vs vs. Remain consistent. *Fixed*

Bio-SiO2: Do you mean biogenic SiO2? Please rephrase. *Bio-SiO$_2$ is commonly used for biogenic SiO$_2$ and is more precise than opal when discussing chemistry and MAR. Researchers often add structural water to the formula of opal, which can change the mass. We discuss our usage at the beginning of the paper, and now use bio-SiO2 throughout the paper.*

Labeling of sites: ODP Sites without "U", IODP Sites with "U". Please correct accordingly. *Corrected*

Line 12: change "late Miocene-Recent" to "late Miocene to recent" *done*

Line 14: Add ODP *done*

Line 15: Delete "The period between" and move "4.5 and 8 Ma" to "LMBB" in the

sentence before. *revised*

Line 20: Change "We use a combined age model: : :" to "We use a combined age model for all sites investigated

by joining: : :". Delete "data from". *The paragraph has been deleted in the revision*

Line 22: Add reference to "At 5 Ma, the sites span from _4_S to _4_N" *We added how we did it; this is a*

*calculation for this paper. A table paleolocations for each drillsite could be added to the supplemental material if*

*needed*.

Line 2: Delete "these" *done*

Line 1: I guess Site 849 is missing in the heading *849 has been added to the heading*.

Lines 11 and 17-18: Remove, because this information is already given on p. 5, l. 2-3. *The paragraph has been*

*revised*

Lines 15-17: Change "Unfortunately, Hagelberg et al (1995) did not publish their CaCO3 estimates for the Leg

138 Sites, so we redid the estimate for Sites 848, 850, and 851 along the revised splices presented here" to "Here

we provide CaCO3 estimates for Sites 848, 850, and 851 along the revised splices". *revised*

Line 2: Remove "unfortunately". *removed*

Line 12-13: Repetition. Delete. *Lines were deleted*

Line 24: Change "CaCO3 MAR records: : :" to "Except for Site U1337, CaCO3 MAR records: : :" and remove

"however" in the following sentence. *revised*

Line 26: Change ": : :that is evidence: : :" to ": : : that is evident: : :" *the relevant phrase was changed to "may be caused by"*

Lines 27-28: Change "Most of the drill sites in this paper have variability in the bulk sediment MAR, but most of the bulk MAR variation is typically derived from changes CaCO3 MAR. Site U1337 is unique: : :" to "Variability in bulk sediment MARs of the sites investigated in this study is typically derived from changes in CaCO3 MAR. However, Site U1337 is unique: : :" *the sentences are reworded as suggested*

Line 2: Change "in addition to" to "our interpretation becomes supported by". *The sentence is changed*

Line 12: Add "that we observe" to "intervals". *added*

Line 23: Change "farther" to "further". *Farther applies to distance, so fits better.*

Lines 25-27: Provide link to figure. *There is no figure of bio-SiO2 MAR at different sites in the paper now. Sites U1335, 848, 850, and 851 do not have an estimate of bio-SiO2, while Site U1337 has an anomalous bio-SiO2 MAR because of the apparent sediment focusing. A figure could be made comparing MARs at Sites U1338 and 849, but this didn't seem needed.*

Lines 29-30: Delete "We infer: : :CaCO3 content". *The sentence was removed.*

Line 20: Delete ":"*deleted*

Line 22: What do you mean by "they"? *Changed to PPLC-3*

Line 9: Change "glacial carbonate cycles" to "glacial-interglacial carbonate cycles". *changed*

Line 31: Delete "for". *Reworded*

Lines 19-20: This information should be part of the figure caption and not of the main text. *The paragraph has been reworded to make the levels of smoothing relevant*

Lines 31-34: Repetition of p. 13, l. 4-8. Please restructure. *The paragraphs have been combined and the*

*redundancy has been eliminated.*

Line 2: Change ": : :XRF scanning data are available, there: : :" to ": : :XRF scanning data are available (i.e., Sites XXX, YYY; ZZZ), there: : :" and remove "at the four drill sites with XRF data" (l. 5-6). *These paragraphs were significantly reworded.*

Lines 30-34: This doesn't work. You already provide a link to Fig 8 earlier. Delete this sentence, but make sure that the information about the smoothed d18O record is given in the figure caption. *fixed*

Lines 2-3: Change "the correlation is still strong when CaCO3:BaSO4 is compared to the smoothed oxygen isotope record" to "that correlation is still strong". *fixed*

Line 12: Add "Site" to 607. *fixed*

Lines 6-9: Delete these sentences. *Sentences have been reworded.*

Lines 1-2: Delete "In this paper: : :work regionally". *This has been reworded*

Lines 4-9: Restructure as follows: "We identified five long- term low CaCO3 intervals within the 7 drill sites we investigated: PPLC-5 (4737-4465 ka), PPLC-4 (3 intervals between 4093 and 2915 ka), PPLC-3 (2 intervals on either side of a CaCO3 high, between 2684 and 2248 ka), PPLC-2 (2135-1685 ka), and PPLC-1 (402-51 ka). With bulk chemical data and the geographic range of the investigated drill sites it is possible to distinguish between dissolution and production as causes of low CaCO3 intervals in the Pliocene-Pleistocene record. We found that PPLC-5 and PPLC-2 result from CaCO3 dilution through diatom production, and the other 3 result from enhanced CaCO3 dissolution." *We used this guidance*

Line 16: Delete "whose 3: : :and 3 Ma". *Section has been reworded*

Line 21: Studies from 2016 are not really "new". Replace "new" by "previous". *Section has been reworded*

Line 7: Remove "Late Miocene Biogenic Bloom". *removed*

Line 7: Change "atandardized" to "standardized". *fixed*

*Reviewer 2 Comments (Daniele Reghellin):*

10 *General response: Many of these comments overlap with that of R1. Distinct issues brought up by R2 are the lack of discussion about the inaccuracies of the GRA CaCO3 estimate, the lack of information about how the CCD depth was calculated by the MAR method, and a need for better discussion about distinguishing a productivity event. We disagree with the comment that CaCO3 cannot be estimated from GRA bulk density, since it has been adequately estimated this way for decades, albeit with about a 6% error when compared to discrete CaCO3*
15 *measurements. We include a better discussion of the CaCO3 MAR gradient method to estimate CCD.*

General Comments

I enjoyed reading and reviewing the Lyle et al. manuscript (MS No.: cp-2018-157). For most of the part, it is a
20 well written and structured manuscript, which presents high resolution sediment properties and compositional records from multiple locations of the eastern equatorial Pacific Ocean. The Results and their interpretation represent a fundamental contribution to the effort of understanding how this region and, more in general low latitude ocean areas, works as the ocean/climate system evolved to modern conditions. The Discussion mainly focuses on five Plio-Pleistocene intervals of low carbonate content. The causes of low carbonate content in the
25 EEP, dilution by biosilica particles versus dissolution of carbonate particles, are here examined using carbonate content, MAR of different sediment components, sedimentation rates, CaCO3:BaSO4 ratios and benthic foraminifera stable isotopes records from multiple locations of the EEP.

The main issues I found in the manuscript mostly concern parts of the Methods and Results and are listed in
30 "Specific comments". The inaccuracies typical of carbonate content estimations from GRA density have not been discussed. It is not clear how CCD depth was calculated from CaCO3 MAR and data are missing. In the Results/

Discussion it is not always clear how intervals of low carbonate content were interpreted as reflecting dilution by biosilica rather than greater carbonate dissolution or vice versa. The Biogenic bloom interval is improperly defined and its proposed end is imprecise. Given this, my overall evaluation is that this manuscript has the potential to be published in Climate of the Past, but I recommend to revise the manuscript according to all the comments presented below before publication.

Specific comments

Abstract. At the beginning of the Abstract, add a couple of sentences introducing the problem(s) at the base of this study and the reasons for this study. Also, add a reference to the study area. There is no reference of the EEP in the whole Abstract. *The abstract was changed by adding 3 sentences to the abstract, including a mention of the eastern equatorial Pacific*

Introduction. As for the Abstract, in the Introduction I couldn't find the unresolved questions that stay at the base of this study. Please add them together with the aims of this study.

*The introduction was revised to better communicate what we are reporting.*

Biogenic bloom. I think it is incorrect to refer to the Biogenic Bloom as "Late Miocene Biogenic Bloom (LMBB)" because this event did not occur only during the Miocene as it ended in the early Pliocene. This is stated in the manuscript several times and is reported in the literature. For example, the studies cited on page 7 line 13 (Farrell et al., 1995, Lyle and Baldauf, 2015) refer to the biogenic bloom as "Biogenic Bloom" or as "late Miocene early Pliocene biogenic bloom". In the paper it is also awkward to read that an early Pliocene interval (e.g. PPLC-5) is part of the LMBB. For example, on page 35 line 4 what is defined as a late Miocene event (LMBB) includes another event from the early Pliocene. It makes little sense to me. I strongly recommend to substitute "late Miocene Biogenic Bloom (LMBB)" with "Biogenic Bloom" or with "late Miocene early Pliocene biogenic bloom" throughout the text, figures, figure captions and tables. Also change all the acronym "LMBB" accordingly, throughout the text, figures, figure captions and tables.

*Most of the Biogenic Bloom falls within the late Miocene, and LMBB is the best succinct description. "Biogenic Bloom" is too generic, and "LMEPBB" is too cumbersome. Similarly, the Miocene warm period is labeled the middle Miocene Climate Optimum (MMCO), even though it actually starts in the early Miocene. The LMBB is made up of multiple high production intervals superimposed upon generally high sedimentation rates with respect to the Pleistocene. PPLC-5 is the last of these high production intervals.*

Data presented. In the text it is not so easy to keep track where the presented data come from (a previous study, this study, etc.) because of the important amount of data. It would make things easier for the reader to add some columns to Table 1 or even adding a new table in which is clearly listed type of data, drill site, data origin and

5    time span of all the data presented in the study.

*Papers where data are located are now included in Table 1; we also rewrote the sections discussing data to make data sources more clear.*

CaCO3 % estimates from GRA density (sections 3.4 and SM 5). There is no mention in the manuscript that

10    carbonate content estimates calculated from GRA measured in EEP sediment have been demonstrated to lead to imprecise carbonate estimations. Reghellin et al. (2013) [Reghellin, D., G. R. Dickens, and J. Backman (2013), The relationship between wet bulk density and carbonate content in sediments from the Eastern Equatorial Pacific, Marine Geology, 344, 41-52] have shown that it is not possible to accurately describe the relationship between carbonate content using a single equation, like it was done in this study. The estimation errors are

15    particularly evident at high CaCO3 content (>60 %; the case of most of the records presented here) because of the wide range of WDB. Fig. SM-4 show that at Site 849 CaCO3 estimation from GRA, XRF and discrete measurements give similar values but still differences are apparent between GRA and discrete measurements in the figure. On the basis of the results presented by Reghellin et al. (2013), new carbonate content records estimated from GRA density cannot be published without fully considering the uncertainties and inaccuracies of

20    the method. In addition, the relationship varies significantly in the EEP from a site to another because of sediment composition differences (carbonate vs biosilica content and, within carbonate components, the type of them) at different locations. In the SM only one power law equation (5) is presented so I have the doubt that it was computed using all discrete CaCO3 and density data from Sites 848, 849, 850 and 851 together. This can potentially bring to even greater errors in the CaCO3 estimates than using different power law equations at

25    different sites. If this is the case, I recommend to recalculate CaCO3 estimates using different power law equations at different sites. Each power law equation should be computed using the data from that particular site only.

*Carbonate in the eastern equatorial Pacific has been shown to be highly correlated with wet bulk density first in a 1979 paper by Larry Mayer ( JSedPet 49,3, 819, now added to the references). Mayer measured grain size and*

30    *fragments as well as carbonate content. He found that carbonate content accounted for 88% of the variance in wet bulk density, while mean grain size accounts for 0.5%, and percent fragments accounted for 0.01%.*

*Sediments lose porosity (gain wet bulk density) as they are buried. If this is not compensated for, there is a trend to higher density with depth and errors for higher CaCO3 values. Most of the observations by Reghellin et al (2013) that GRA is unsuitable for estimating CaCO3 probably arise because they did not decompact the GRA density. As for the CaCO3 estimate being less certain at higher CaCO3 percentages, see the supplemental material and see the plot by Hagelberg et al (1995, below), who found, as we did, that there was no trend of worse predictability with higher CaCO3 contents. Hagelberg et al (1995), used essentially the same method as we did (we used a slightly different decompaction curve), and got root mean square (RMS) errors averaging 7% for estimated CaCO3 % by GRA for all 8 sites, similar to the 6% we got for Site 849 versus ~4% for XRF estimates of CaCO3 %. What we found, as discussed in the text are similar errors in the GRA method to Hagelberg et al. (1995). Figure SM-4 shows that the trends of discrete, XRF, and GRA estimates of CaCO3 % are the same and reasonably fit the discrete measurements.*

[Figure]

Figure 4. Scatter plots of predicted (from GRAPE) vs. measured %CaCO$_3$ from the top 6 m.y. of Sites 846 through 853.

MAR data (section 3.5). I tried to calculate some MAR for a few sediment components using the equation (6), with dry bulk density, sedimentation rates and CaCO3 % data in the SM tables and the results were different from corresponding MAR data listed in your tables. Please verify if data presented in the tables are correct.

*The data in the MAR tables are correct. I can only assume that R2 tried to calculate via individual samples, and not using the smoothed data found in tables SM-26 to SM-32. As explained in the text, the smoothing was done to minimize spikes caused by errors in sedimentation rate and individual bulk chemical analyses. Applying smoothed sedimentation rates to the unsmoothed data will result in erroneous results.*

Biogenic bloom end. On page 7 line 16 is stated "is easily observed by the change in slope of the age-depth curves at about 4400 ka (Fig. 2)". To me is evident that the most significant and clear change in the age-depth curves of Fig. 2 is a bit to the right of where the grey line is placed, which is at about 4600 ka. This is particularly evident in curves of Site U848, U1338, U851, U850 and U849. I would then strongly recommend to: i. move the end of the biogenic bloom line at 4600 ka in Fig. 2; ii. move green arrow tip to 4600 ka; iii. recalculate average sedimentation rates for the periods 0-4600 ka and 4600-7200 ka in Fig. 2; iv. modify all figures, figure captions and manuscript text according to the above reasoning.

*This is now Fig 6 in the revised manuscript. We found that the end of PPLC-5, marking the last productivity interval in the LMBB, occurred at 4465 Ma. The figure is meant to be illustrative and the average sedimentation rates will change little if the dividing line is moved to 4600 ka. The figure was not used to determine the end of the LMBB since examination of sedimentation rates is a more useful measure. The R2 eyeball estimate of 4600 ka for the bend is probably older than the actual sedimentation rate changes. We propose to leave the figure alone.*

Sediment focusing (page 10 line 6). "Surprisingly, the sediment: : : : : :strongly affect the CaCO3 % profile" isn't this because of moderate currents strength preferentially removes light-fine sediments as you say on lines 1 and 2? If this is the case it makes sense to me that sedimentation rates increase, CaCO3 % decreases but leaving the CaCO3 % profiles unaltered. And CaCO3 MAR would increase because of higher sedimentation rates, even if dry density decreases because of the input of fine sediments. This is not anomalous to me.

*We agree with this reasoning and it matches what we were trying to say in the text. The work we have done on sediment focusing (Lyle et al., 2014; Lovely et al., 2017) suggests that fine-grained materials are preferentially moved except at high current speeds where all the sediment moves.*

Dilution vs dissolution. In the Results (sections 5.3-5.5) it is often missing a clear explanation of how it was distinguished between carbonate dilution by biosilica and carbonate dissolution. Section 5.3. The first paragraph (page 10 lines 12-19) is difficult to follow and needs rewriting because it is not clear how you interpreted some intervals (i.e. PPLC-5 and PPLC-2) as high production rather than high carbonate dissolution. Is it because carbonate% is low at on equator sites and not as much at off equator sites? or is it because of low carbonate% and relatively high carbonate MAR at on equator sites? The CaCO3:BaSO4 used to estimate carbonate dissolution is also very low at PPLC-2 at all sites (Fig. 7). Couldn't it be that high production also generated greater carbonate dissolution? How is it possible to completely exclude carbonate dissolution in this interval? I recommend to

clearly state in which way it is possible to distinguish between dilution and dissolution. Section 5.4. Lines 14-15: it is not clear how data at PPLC-4 indicate dissolution rather dilution by biosilica. Provide clear explanation. Section 5.5. Here it is easier to follow your rationale because you introduce the CaCO3:BaSO4 proxy. I strongly recommend to add references to this proxy in the interpretation of the other PPLC intervals (so in sections 5.3 and 5.4). It would make your interpretation of low carbonate intervals much stronger and easier to follow by the reader.

*These are now sections 4.3 to 4.5. We have revised the section to make it more clear. As explained in the paragraph in question a combination of MAR and percentage data marks a productivity interval. The information at an individual drill site is supported by changes along a set of drill sites occupying a latitudinal transect away from the equator since upwelling is focused at the equator. Reghellin is correct that production and dissolution are not mutually exclusive, and we agree that CaCO3:BaSO4 is low in the PPLC-2 interval. We will add something to the text about CaCO3:BaSO4 for other PPLC intervals.*

CCD depth estimate from CaCO3 MAR (section 6.1). It is not clear how MAR CCD were calculated. From reasoning in 6.1 it seems that you have calculated MAR CCD from CaCO3 MAR using equation (1) in page 13, but in that equation there is no MAR CCD. Clarify this. I could not find the data used to calculate MAR CCD anywhere in the manuscript nor in SM. I strongly suggest to add a table listing all data used to make curves in Fig. 5 panel b. While reading section 6.1 I got the feeling that the estimation of CCD depth from CaCO3 MAR is a weak approach. This is because you state that i) "local sediment anomalies: : : : :cause significant noise to this approach" ii) "weaker signal in the Pliocene and Miocene is harder to distinguish from noise", iii) the CCD estimate "suffers from the noise resulting from building the trend with records from only 2 sites" and iv) "minor errors in correlation: : : : :are magnified in the CCD estimate". Plus, it is not clear how you calculated CCD depth from CaCO3 MAR.

Also, I see there are significant differences between the CCD depth record (Fig. 5 panel b) and the equatorial Pacific CCD of Pälike et al. (2012). The latter ranges between about 4200 and 4700 mbsl whereas yours mostly between 4300 and 5500 mbsl. How can you explain this difference? It seems to me that all the issues of the method are strongly affecting the CCD estimates. Isn't it enough to speculate on carbonate preservation (dissolution) over time by using the CaCO3:BaSO4, CaCO3 MAR, CaCO3 % and sedimentation rates? In my opinion the message emerging from this part of the Discussion would be much stronger without the CCD depth estimate.

*We redid this section to better explain how the MAR extrapolation of the CCD is done in both the text and in the*

*supplementary material.*

Page 17 lines 22-24. Add that evidence of early Pliocene higher SST and lower biogenic production compared to during the biogenic bloom comes also from bulk sediment stable isotopes and sedimentation rates records from Leg 138 on-equator sites and from Site 573 and U1338 (see Shackleton and Hall., 1995 and Reghellin et al., 2015) [Reghellin, D., H. K. Coxall, G. R. Dickens, and J. Backman (2015), Carbon and oxygen isotopes of bulk carbonate in sediment deposited beneath the eastern equatorial Pacific over the last 8 million years, Paleoceanography, 30, doi:10.1002/2015PA002825].

*This will be added.*

- Page 18 lines 5-6. Add explanation of how the "expansion of an Antarctic source: : : : : :and Pacific" can cause opposite changes in the d13C records from the Atlantic and the Pacific

*The Antarctic mixes all subsurface flows to get an average d13C. If the Atlantic outflow is displaced to a somewhat shallower depth (as happened at the LGM), Atlantic benthic d13C goes down, while the Pacific d13C may stay the same or go upward, depending on how much high d13C water is generated in the Atlantic. We will add better explanation to the discussion.*

Technical corrections

Page 1 Line 12: define "XRF" Line 13: define "IODP" and "ODP" Line 23: define "DIC" *done*

Page 2 Line 2: add "Reghellin et al., 2013" to references. *done*

Line 25: define "DIC" *done*

Line 27: add references *added*

Line 28: define "NADW" *done*

Line 34: substitute square brackets with round brackets. *done*

Page 3 Lines 4-7: move this sentence to Results or Discussion Lines 13-15: add references Line 26: define "XRF" Line 31: define "IODP" *Sentence removed; XRF, IODP,ODP defined in abstract*

Line 32: define "ODP" and "GRA" *done*

Page 4 Line 2: change "3_S" to "3.0_S" Line 11: define "ccsf". Also, "ccsf" is in the textnsometimes written as lowercase and sometimes as uppercase. Modify to be consistent *defined, and consistently caps now*

Page 5 Line 11: add explanation of the criteria used to choose the base or master drill site, *added*

Page 6 Line 5: define "ICP-MS" Line 14: add "Reghellin et al., 2013" to references *done*

Page 7 Lines 16-17: substitute "4400 ka" with "4600 ka" Line 20: define "DSDP" *only if we change the figure. DSDP has now been defined*

Page 8 Line 3: late Miocene carbonate crash here defined from 11 to 8 Ma and on page 7 line 11 from 10 to 8 Ma. Change time periods to be consistent *Changed*

5 Line 5: in References list there are two Pälike et al., 2010b. To which one do you refer? References list *we realized that all references that we used could be directed to the Exp 320/321 summary, so eliminated the reference to the entire volume.*

Page 9 Lines 3-8: the same concept is repeated in these two sentences. Keep one and delete the other. I would also add low carbonate production as a cause for low CaCO3 intervals. *Text has been revised*

10 Page 10 Line 26: add "(Figure 3)" after "at Site U1338" *done*

Page 11 Line 1: define "MIS" Line 14: substitute "lowest record" with "panel b" Lines

15-19: these are speculations. Move to Discussion Line 22: substitute "bottom" with "panel b" *In revision we have defined the CaCO3/BaSO4 ratio, so we refer to observations*

Page 12 Lines 27-30: add references *this was adequately referenced in the introduction*

15 Page 13 Line 25: is (1) a novel equation? If so state it otherwise provide reference *it is neither novel nor taken from another reference.*

Page 15 Line 5: how about PPLC-2? I see that also during this time period the CaCO3:BaSO4 is very low at XRF sites Line 32: specify the type of smoothing *The smoothing results from the stacking; noise cancels out because it is uncorrelated. We add a sentence about dissolution in PPLC-2*

20 Page 16 Line 10: define "AABW" *done*

Page 17 Line 5: substitute "implies" with "would imply" Line 11: define "NCW" Line 15: define "SST" *changed*

Page 18 Line 31: add "early Pliocene" after "late Miocene" Line 32: add "Reghellin et al., 2015" *early Pliocene added; did not add Reghellin reference*

Page 19 Line 34: define "CAS" *defined*

25 Page 27 There are two Pälike et al., 2010b. Fix*fixed*

Page 31 Table 1: change "length (ka) of dated record" to "length of dated record (ka)". *That column has been removed on advice of R1*

Can you specify what is meant with "Data available"? See comment in "Specific comments" *references have been made*

30 Page 32 Specify origin of background map. In the map are present many light grey dots which (drill sites locations?) that are not labelled. Add a label or remove the dots *fixed*

Page 33 See comment "Biogenic bloom end" in Specific comments section Figure 3:subscript 3 in "CaCO3" in panel a scales Substitute "4400 ka" with "4600 ka" in figureand figure caption In figure caption, define what is the grey vertical line *now Fig 6 see our comments about moving the line.*

Page 35 What are darker curves on panel a? is it a kind of smoothing of actual data record? Clarify in figure caption Line 4: see "biogenic bloom" comment in "Specific comments" *fixed; it is a smooth*

Page 36 Can you revers y axis scale on panel b? With shallower depths at the top it is difficult to follow lines 5-6: see "CCD depth estimate from CaCO3 MAR" comment *done*

in "Specific comments" line 6: specify which curve is the 50 kyr smoothing (blue line?) and which one is the 750 kyr smoothing (bold blue line?) *added to caption*

Page 37 Please add a sentence in figure caption about alignment of d18O glacial intervals and high opal MAR*added*

Page 39 Specify in figure caption what the dark red line on the stack isotope curve is *added*

Page 40 Specify in figure caption what are dark color lines *added*

SM Page 1 Lines 26-27: given what is written here I do not understand why in the

captions of Tables SM-1 to SM-4 (ODP sites) you refer to CSF and CCSF depths and in captions of Tables SM-5 to SM-7 (IODP sites) you refer to mbsf and rmcd depths. Please correct depth scales in all supplemental tables or clarify*This was clarified and was intended to reflect the change in nomenclature that is needed to*

SM Page 2 Line 16: define "IMPH"*clarified*

SM page 5 Line 24: substitute "since they have partially: : :" with "because they have partially: : :" Line 25: define "APC" Line 26: define "XCB" *done*

SM page 6 Lines 30-31: describe method used to obtain GRA estimates *expanded*

SM page 7 Lines 8-9: can you state where the "discrete CaCO3 data" come from? And can you clarify with which data the power law estimate (5) was computed? Line 23: specify the unit of "Xe" Line 24: change "grams" to "g/cm3" *fixed. Xe is unitless (grams/grams).*

SM page 8 Lines 19-20: Can you provide a possible cause of this sediment accumulation/erosion feature at Site U1337? Lines 22-23: "Site U1337: : : : : :significantly elevated relative to the CaCO3 MAR" add reference to Fig. 5 *fixed*

SM page 12 On panel d there are two marks for Site 572 location. Fix SM page 14 Change "Figure SM-1" to "Figure SM-3" *Gray point is the site survey piston core RR0603-4JC*

SM page 15 Change "Figure SM-2" to "Figure SM-4" *fixed*

Daniele Reghellin

**References cited**

Hagelberg, T. K., Pisias, N. G., Mayer, L. A., Shackleton, N. J., and Mix, A. C.: Spatial and temporal variability of late Neogene equatorial Pacific Carbonate: Leg 138. . In: *Proceedings of the Ocean Drilling Program, Scientific Results*, 138, 321-336, 1995.

Lovely, M. R., Marcantonio, F., Lyle, M., Ibrahim, R., Hertzberg, J. E. and Schmidt, M. W.: Sediment redistribution and grainsize effects on $^{230}$Th-normalized mass accumulation rates and focusing factors in the Panama Basin, *Earth and Planetary Science Letters*, 480, 107-120, 2017.

Lyle, M., Marcantonio, F., Moore, W. S., Murray, R. W., Huh, C.-A., Finney, B. P., Murray, D. W. and Mix, A. C.: Sediment size fractionation and sediment focusing in the equatorial Pacific: effect on $^{230}$Th normalization and paleoflux measurements, *Paleoceanography*, 29, 747-763, doi:10.1002/2014PA002616, 2014.

Mayer, L. A.: Deep-sea carbonates: acoustic, physical, and stratigraphic properties, Journal of Sedimentary Petrology, 49, 819-836, 1979.

Mix, A. C., Pisias, N. G., Rugh, W., Wilson, J., Morey, A., and Hagelberg, T. K.: Benthic foraminifer stable isotope record from Site 849 (0-5 Ma): local and global climate change. . In: *Proceedings of the Ocean Drilling Program, Scientific Results*, 138, 371-412, 1995.

Pälike, H. et al.: A Cenozoic record of the equatorial Pacific carbonate compensation depth, *Nature*, 488, 609-614, doi:10.1038/nature11360, 2012.

Reghellin, D., Dickens, G. R. and Backman, J.: The relationship between wet bulk density and carbonate content in sediments from the eastern equatorial Pacific, *Marine Geology*, 344, 41-52, 2013.

---

## Author Response (AR2)

Response to second review of Lyle et al, submitted to Climate of the Past in Oct 2018. Reviewers comments are in black while our responses are in blue. A markup version of the revised manuscript is attached to this response.

This is the second review of the manuscript "Late Miocene to Recent High Resolution Eastern Equatorial Pacific Carbonate Records: Stratigraphy linked by dissolution and paleoproductivity" from Mitchell Lyle et al. The new manuscript version is well structured and reads much better compared to the previous version, and I was able to follow it more easily. Most of the concerns raised in the last revision have been addressed by the authors, e.g., the aim of their study is now clearly stated in both the abstract and the introduction. I have a small number of additional comments that could help to further improve the manuscript and a few very minor technical corrections.

Specific comments

Introduction:
The introduction section is quite long. I was wondering whether it would be better to give a short introduction that defines the aim of your work and then present a second chapter that gives all the background information on the CCD evolution and CaCO3 deposition/dissolution in the east Pacific over the Cenozoic.
We divided the introduction into 2 sections, the first outlining our objectives and their relationship to PEAT drilling, and the second giving the background of studies of the CCD.

p. 4, lines 27-28:
The amount of carbonate production should only influence the clay percentages but not the absolute value of clay deposition.
We have revised the sentence to make the meaning clear. Higher dissolution and lower carbonate production increase the relative clay content. In addition, there is higher clay deposition to the north of the equator because of the position of the ITCZ and rainout of dust there.

p. 6, line 14:
The stable isotope record from Mix et al. (1995) goes back to 5 Ma. Why is only the 0–3.6 Ma interval used for the age model you generate? I think the explanation is given in line 29, but this information should be presented earlier.
A sentence was added to the topic paragraph to explain why the join was made at 3.6 Ma

p. 13, line 27:
Is there any evidence that a significant amount of deep waters from the North Pacific bathed the studied sites during low CaCO3 intervals? Typically, it is argued that deep waters in the central east Pacific are fed by deep waters deriving from the high southern latitudes.
Deep circulation in the Pacific is more 3-dimensional than in the Atlantic. Circumpolar deep water flows north in the western Pacific with a small diversion going NE around Hawaii.  The return flow from the North Pacific Basin travels south in the eastern Pacific,

along the western flank of the East Pacific Rise. We mentioned this flow in the introduction but reiterate the flow pattern here. One of the best visualizations is along the WOCE P04 line, E-W at 10°N (http://whp-atlas.ucsd.edu/pacific_index.html) which documents the higher nutrients along the west flank of the East Pacific Rise from North Pacific outflow.

p. 14, line 25:
I imagine that the benthic/planktic ratio of foraminifera at Site U1338 shows the same pattern that derives from the CaCO3 to bio-BaSO4 ratio from the same site as described in the paragraph above? Please make this clear in the text.
We added a sentence to the effect that the benthic/planktic ratio could be used to assess dissolution as well, although foram counts were not done in this paper.

p. 17, line 19:
A few words should be spend on the timing of CAS closure (cutoff of deep-water exchange in the mid-Miocene; surface-water exchange until the late Pliocene/early Pleistocene; Stephs et al., 2010, Groeneveld et al., 2014, Sepulchre et al., 2014) and its impact on east Pacific primary productivity typically assumed (e.g., Schneider & Schmittner 2006).
We added two sentences documenting the CAS closure.

Main text:
While reading the manuscript, I had the impressions that there are several repetitions particularly in the discussion section. Please check and avoid repetitions wherever possible since the manuscript is quite long.
We went through the discussion section again and streamlined it.

Technical corrections
p. 2, line 9: Delete "that". English requires "that" in the sentence as written.
p. 2, line 12: Unclear which processes are meant. "Surface and abyssal" added
p. 1, line 15: Introduce CCD. Added a reference to CCD at this point
p. 3, line 28: "had to be" instead of "was to be"?.The sentence was revised for clarity.
p. 3, line 32: Delete "rapid". done
p. 4, line 2: Not clear what you mean with "above the CCD from Pälike et al.".We eliminate the Pälike et al reference, since the sites we were studying are all above the CCD.
p. 4, lines 19 & 24: "Seven" vs "7". Remain consistent throughout the manuscript. All references to the number of drillsites are now listed as "7"
p. 7, line 3: Delete "data". "Data" was eliminated.
p. 7, lines 2-11: There are some repetitions in this paragraph, please reorganize. We have significantly reorganized section 3.3 to eliminate redundancies.
p. 13, lines 9.10: Add references. I referenced Haug et al. (2005) and added the reference.

[revised manuscript text omitted]

---

## Author Response (AR3)

Authors Response, final submission of cp-2018-157

We submit the final file uploads. We include a clean pdf document of the whole manuscript and the MS Word .docx version. We also include pdf versions of all figures that were produced using Adobe Illustrator. There were no text revisions from the editor to fix.

We have added *Code/Data Availability, Author Contributions, and Competing interests* to the text, and have copied the excel files into MS Word tables in the text rather than using a screen grab. If you would like the original excel files, please let us know. We have checked the figures to make certain that the fonts were embedded.